# CONSISTENT COUNTERFACTUALS FOR DEEP MODELS

**Emily Black,**[*] **Zifan Wang,**[*] **Anupam Datta, Matt Fredrikson**
{`emilybla, zifan, danupam, mfredrik`} `@cmu.edu`
Carnegie Mellon University

## ABSTRACT

Counterfactual examples are one of the most commonly-cited methods for explaining the predictions of machine learning models in key areas such as finance and medical diagnosis. Counterfactuals are often discussed under the assumption that the model on which they will be used is static, but in deployment models may be periodically retrained or fine-tuned. This paper studies the consistency of model prediction on counterfactual examples in deep networks under small changes to initial training conditions, such as weight initialization and leave-one-out variations in data, as often occurs during model deployment. We demonstrate experimentally that counterfactual examples for deep models are often inconsistent across such small changes, and that increasing the cost of the counterfactual, a stability-enhancing mitigation suggested by prior work in the context of simpler models, is not a reliable heuristic in deep networks. Rather, our analysis shows that a model's Lipschitz continuity around the counterfactual, along with confidence of its prediction, is key to its consistency across related models. To this end, we propose Stable Neighbor Search[1] as a way to generate more consistent counterfactual explanations, and illustrate the effectiveness of this approach on several benchmark datasets.

## 1 INTRODUCTION

Deep Networks are increasingly being integrated into decision-making processes which require explanations during model deployment, from medical diagnosis to credit risk analysis (Bakator & Radosav, 2018; et. al, 2017; Liu et al., 2014; Sun et al., 2016; De Fauw et al., 2018; Babaev et al., 2019; Addo et al., 2018; Balasubramanian et al., 2018; Wang & Xu, 2018). *Counterfactual examples* (Wachter et al., 2018; Van Looveren & Klaise, 2019; Mahajan et al., 2019; Verma et al., 2020; Laugel et al., 2018; Keane & Smyth, 2020; Ustun et al., 2019; Sharma et al., 2019; Poyiadzi et al., 2020; Karimi et al., 2020; Pawelczyk et al., 2020a) are often put forth as a simple and intuitive method of explaining decisions in such high-stakes contexts (Mc Grath et al., 2018; Yang et al., 2020). A counterfactual example for an input $x$ is a related point $x'$ that produces a desired outcome $y'$ from a model. Intuitively, these explanations are intended to answer the question, "Why did point $x$ not receive outcome $y'$?" either to give instructions for *recourse*, i.e. how an individual can change their behavior to get a different model outcome, or as a check to ensure a model's decision is well-justified (Ustun et al., 2019). Counterfactual examples are particularly popular in legal and business contexts, as they may offer a way to comply with regulations in the United States and Europe requiring explanations on high-stakes decisions (e.g. Fair Credit Reporting Act (FCRA) and General Data Protection Regulation (GDPR, 2016)), while revealing little information about the underlying model (Barocas et al., 2020; Mc Grath et al., 2018).

Counterfactual examples are often viewed under the assumption that the decision system on which they will be used is static: that is, the model that *creates* the explanation will be the *same* model to which, e.g. a loan applicant soliciting recourse re-applies (Barocas et al., 2020). However, during real model deployments in high-stakes situations, models are not constant through time: there are often retrainings due to small dataset updates, or fine-tunings to ensure consistent good behavior (Merchant, 2020; pwc, 2020). Thus, in order for counterfactuals to be usable in practice, they must return the same desired outcome not only for the model that generates them, but for similar models created during deployment.

This paper investigates the consistency of model predictions on counterfactual examples between deep models with seemingly inconsequential differences, i.e. random seed and one-point changes in the training set.

---

[*]Equal Contribution
[1]Implementation is available at `https://github.com/zifanw/consistency`

We demonstrate that some of the most common methods generating counterfactuals in deep models either are highly inconsistent between models or very costly in terms of distance from the original input. Recent work that has investigated this problem in simpler models (Pawelczyk et al., 2020b) has pointed to increasing counterfactual cost, i.e. the distance between an input point and its counterfactual, as a method of increasing consistency. We show that while higher than *minimal* cost is necessary to achieve a stable counterfactual, cost alone is not a reliable signal to guide the search for stable counterfactuals in deep models (Section 3).

Instead, we show that a model's Lipschitz continuity and confidence around the counterfactual is a more reliable indicator of the counterfactual's stability. Intuitively, this is due to the fact that these factors bound the extent of a models local decision boundaries will change across fine-tunings, which we prove in Section 4. Following this result, we introduce *Stable Neighbor Search* (SNS), which finds counterfactuals by searching for high-confidence points with small Lipschitz constants in the generating model (Section 4). Finally, we empirically demonstrate that SNS generates consistent counterfactuals while maintaining a low cost relative to other methods over several tabular datasets, e.g. Seizure and German Credit from UCI database (Dua & Karra Taniskidou, 2017), in Section 5.

In summary, our main contributions are: 1) we demonstrate that common counterfactual explanations can have low consistency across nearby *deep* models, and that cost is an insufficient signal to find consistent counterfactuals (Theorem. 1); 2) to navigate this cost-consistency tradeoff, we prove that counterfactual examples in a neighborhood where the network has a small local Lipschitz constant are more consistent across changes to the last layer of weights, which suggests that such points are more stable across small changes in the training environment (Theorem. 2) ; 3) leveraging this result, we propose SNS as a way to generate consistent counterfactual explanations (Def. 5); 4) we empirically demonstrate the effectiveness of SNS in generating consistent and low-cost counterfactual explanations (Table 1). More broadly, this paper further develops a connection between the geometry of deep models and the consistency of counterfactual examples. When considered alongside related findings that focus on attribution methods, our work adds to the perspective that *good explanations require good models to begin with* (Croce et al., 2019; Wang et al., 2020; Dombrowski et al., 2019; Simonyan et al., 2013; Sundararajan et al., 2017).

## 2 BACKGROUND

**Notation.** We begin with notation, preliminaries, and definitions. Let $F(\mathbf{x};\theta) = \mathrm{argmax}_i f_i(\mathbf{x};\theta)$ be a deep network where $f_i$ denotes the logit output for the $i$-th class and $\theta$ is the vector of trainable parameters. If $F(\mathbf{x};\theta) \in \{0,1\}$, there is only one logit output so we write $f$. Throughout the paper we assume $F$ is piece-wise linear such that all the activation functions are ReLUs. We use $||\mathbf{x}||_p$ to denote the $\ell_p$ norm of a vector $\mathbf{x}$ and $B_p(\mathbf{x},\epsilon) \overset{\text{def}}{=} \{\mathbf{x}' | ||\mathbf{x}'-\mathbf{x}||_p \leq \epsilon, \mathbf{x}' \in \mathbb{R}^d\}$ to denote a norm-bounded ball around $\mathbf{x}$.

**Counterfactual Examples.** We introduce some general notation to unify the definition of a counterfactual example across various approaches with differing desiderata. In the most general sense, a counterfactual example for an input $\mathbf{x}$ is an example $\mathbf{x}_c$ that receives the different, often targeted, prediction while minimizing a user-defined *quantity of interest* (QoI) (see Def. 1): for example, a counterfactual explanation for a rejected loan application is a related hypothetical application that was accepted. We refer to the point $\mathbf{x}$ requiring a counterfactual example the *origin point* or *input* interchangeably. We note that there is a different definition of "counterfactual" widely used in the causality literature, where a counterfactual is given by an intervention on a causal model that is assumed to generate data observations (Pearl, 2009). This is a case of overlapping terminology, and is orthogonal to this work. We do not consider causality in this paper.

**Definition 1** (Counterfactual Example). *Given a model $F(\mathbf{x})$, an input $\mathbf{x}$, a desired outcome class $c \neq F(\mathbf{x};\theta)$ , and a user-defined quantity of interest $q$, a counterfactual example $\mathbf{x}_c$ for $\mathbf{x}$ is defined as $\mathbf{x}_c \overset{\text{def}}{=} \mathrm{argmin}_{F(\mathbf{x}';\theta)=c} q(\mathbf{x}',\mathbf{x})$ where the* cost *of $\mathbf{x}_c$ is defined as $||\mathbf{x}-\mathbf{x}_c||_p$.*

The majority of counterfactual generation algorithms minimize of $q_{\text{low}}(\mathbf{x},\mathbf{x}') \overset{\text{def}}{=} ||\mathbf{x}-\mathbf{x}'||_p$, potentially along with some constraints, to encourage low-cost counterfactuals (Wachter et al., 2018). Some common variations include ensuring that counterfactuals are attainable, i.e. not changing features that cannot be changed (e.g. sex, age) due to domain constraints (Ustun et al., 2019; Lash et al., 2017), ensuring sparsity, so that fewer features are changed (Dandl et al., 2020; Guidotti et al., 2018), or incorporating user preferences into what features can be changed (Mahajan et al., 2019). Alternatively, a somewhat distinct line of work (Pawelczyk et al., 2020a; Van Looveren & Klaise, 2019; Joshi et al., 2019) also adds constraint to ensure that counterfactuals come from the data manifold. Other works still integrate causal validity into counterfactual search (Karimi et al., 2020), or generate multiple counterfactuals at once (Mothilal et al., 2020).

We focus our analysis on the first two approaches, which we denote *minimum-cost* and *data-support* counterfactuals. We make this choice as the causal and distributional assumptions used in other counterfactual generation methods referenced are specific to a given application domain, whereas our focus is on the general properties of counterfactuals across domains. Specifically, we evaluate our results on minimum-cost counterfactuals introduced by Wachter et al. (2018), and data-support counterfactuals from Pawelczyk et al. (2020a), and Van Looveren & Klaise (2019). We give the full descriptions of these approaches in Sec. 5.

**Counterfactual Consistency.** Given two models $F(\mathbf{x};\theta_1)$ and $F(\mathbf{x};\theta_2)$, a counterfactual example $\mathbf{x}_c$ for $F(\mathbf{x};\theta_1)$ is consistent with respect to $F(\mathbf{x};\theta_2)$ means $F(\mathbf{x}_c;\theta_1) = F(\mathbf{x}_c;\theta_2)$. Following Pawelczyk et al. (2020b), we define the *Invalidation Rate* for counterfactuals in Def. 2.

**Definition 2** (Invalidation Rate). *Suppose $\mathbf{x}_c$ is a counterfactual example for $\mathbf{x}$ found in a model $F(\mathbf{x};\theta)$, we define the invalidation rate* IV($\mathbf{x}_c$,$\Theta$) *of $\mathbf{x}_c$ with respect to a distribution $\Theta$ of trainable parameters as* IV$(\mathbf{x}_c,\Theta) \overset{\text{def}}{=} \mathbb{E}_{\theta' \sim \Theta}\mathbb{I}[F(\mathbf{x}_c;\theta') \neq F(\mathbf{x}_c;\theta)]$.

Throughout this paper, we will call the model $F(\mathbf{x};\theta)$ that creates the counterfactual the *generating* or *base* model. Recent work has investigated the consistency of counterfactual examples across similar linear and random forest models (Pawelczyk et al., 2020b). We study the invalidation rate with respect to the distribution $\Theta$ introduced by arbitrary differences in the training environment, such as random initialization and one-point difference in the training dataset. We also assume $F(\mathbf{x};\theta')$ uses the same set of hyper-parameters as chosen for $F(\mathbf{x};\theta)$, e.g. the number of epochs, the optimizer, the learning rate scheduling, loss functions, etc.

## 3 COUNTERFACTUAL INVALIDATION IN DEEP MODELS

As we demonstrate in more detail in Section 5, counterfactual invalidation is a problem in deep networks on real data: empirically, we find that counterfactuals produce inconsistent outcomes in duplicitous deep models up to 94% of the time.

Previous work investigating the problem of counterfactual invalidation (Pawelczyk et al., 2020b; Rawal et al., 2021), has pointed to increasing counterfactual cost as a potential mitigation strategy. In particular, they prove that higher cost counterfactuals will lead to lower invalidation rates in linear models in expectation (Rawal et al., 2021), and demonstrate their relationship in a broader class of well-calibrated models (Pawelczyk et al., 2020b). While this insight provides interesting challenge to the perspective that low cost counterfactuals should be preferred, we show that cost alone is insufficient to determine which counterfactual has a greater chance of being consistent at generation time in deep models.

The intuition that a larger distance between input and counterfactual will lead to lower invalidation rests on the assumption that the distance between a point $x$ and a counterfactual $x_c$ is indicative of the distance from $x_c$ to the decision boundary, with a higher distance making $x_c$'s prediction more stable under perturbations to that boundary. This holds well in a linear model, where there is only one boundary (Rawal et al., 2021).

However, in the complex decision boundaries of deep networks, going farther away from a point across the *nearest* boundary may lead to being closer to a *different* boundary. We prove that this holds even for a one-hidden-layer network by Theorem 1. This observation shows that a counterfactual example that is farther from its origin point may be equally susceptible to invalidation as one closer to it. In fact, we show that the *only* models where $\ell_p$ cost is universally a good heuristic for distance from a decision boundary, and therefore by the reasoning above, consistency, are linear models (Lemma 1).

**Theorem 1.** *Suppose that $H_1, H_2$ are decision boundaries in a piecewise-linear network $F(\mathbf{x}) = sign\{w_1^\top ReLU(W_0\mathbf{x})\}$, and let $\mathbf{x}$ be an arbitrary point in its domain. If the projections of $\mathbf{x}$ onto the corresponding halfspace constraints of $H_1,H_2$ are on $H_1$ and $H_2$, then there exists a point $\mathbf{x}'$ such that:*

$$1)\; d(\mathbf{x}',H_2) = 0 \qquad 2)\; d(\mathbf{x}',H_2) < d(\mathbf{x},H_2) \qquad 3)\; d(\mathbf{x},H_1) \leq d(\mathbf{x}',H_1)$$

*where $d(\mathbf{x},H_*)$ denotes the distance between $\mathbf{x}$ and the nearest point on a boundary $H_*$.*

**Lemma 1.** *Let $H_1,H_2,F$ and $\mathbf{x}$ be defined as in Theorem 1. If the projections of $\mathbf{x}$ onto the corresponding halfspace constraints of $H_1,H_2$ are on $H_1$ and $H_2$, but there* does not *exist a point $\mathbf{x}'$ satisfying* (2) *and* (3) *from Theorem 1, then $H_1 = H_2$.*

Figure 1 illustrates the geometric intuition behind these results. The shaded regions of 1b correspond to two decision surfaces trained from different random seeds on the data in (a). The lighter gray region denotes where the models disagree, whereas the black and white regions denote agreement. Observe

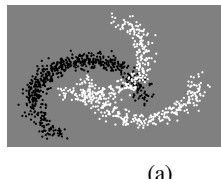 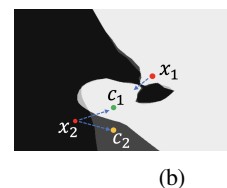 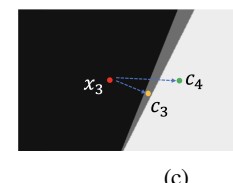

(a)          (b)          (c)

Figure 1: Illustration of the boundary change in a deep model (b) and a linear model (c) for a 2D dataset (a) when changing the seed for random initialization during the training. Shaded regions correspond to the area when two deep models in (b) (or two linear models in (c)) make different predictions.

that counterfactuals equally far from a decision boundary may have different invalidation behavior, as demonstrated by the counterfactuals $c_1$ and $c_2$ for the point $x_2$. Also note that as shown with $x_1$, being far away from one boundary may lead one to cross another one in deep models. However, for two linear models shown in Fig. 1c, being far away from the boundary is indeed a good indicator or being consistent.

The discussion so far has demonstrated that there is not a strong theoretical relationship between cost and invalidation in deep models. In Section 5, we test this claim on real data, and show that higher-cost counterfactuals can have *higher* invalidation rates than their lower-cost relatives (c.f. Table 1). Further, we show that the coefficient of determination ($R^2$) between cost and invalidation rate is very small (with all but one around 0.05). Thus, while cost and invalidation are certainly related—for example, it may be necessary for a stable counterfactual to be more costly than the *minimum* point across the boundary—cost alone is not enough to determine which one will be the most consistent in deep models.

## 4    TOWARDS CONSISTENT COUNTERFACTUALS

In this section, we demonstrate that the Lipschitz continuity (Def. 3) of a neighborhood around a counterfactual can be leveraged to characterize the consistency of counterfactual explanations under changes to the network's parameters (Section 4.2). Our main supporting result is given in Theorem 2, which shows that a model's Lipschitz constant in a neighborhood around a $x_c$ together with the confidence of its prediction on $x_c$ serve as a proxy for the difficulty of invalidating $x_c$. We further discuss insights from these analytical results and introduce an effective approach, *Stable Neighbor Search*, to improve the consistency of counterfactual explanations (Section 4.3). Unless otherwise noted, this section assumes all norms are $\ell_2$.

**Definition 3** (Lipschitz Continuity). *A continuous and differentiable function $h : S \to \mathbb{R}^m$ is $K$-Lipschitz continuous iff $\forall \mathbf{x}' \in S, ||h(\mathbf{x}') - h(\mathbf{x})|| \leq K ||\mathbf{x}' - \mathbf{x}||$. We write $h$ is $K$-Lipschitz in $S$.*

### 4.1    RELU DECISION BOUNDARIES AND DISTRIBUTIONAL INFLUENCE

We analyze the differences between models with changes such as random initialization by studying the differences that arise in their decision boundaries. In order to capture information about the decision boundaries in analytical form, we introduce *distributional influence*: a method of using a model's gradients to gather information its local decision boundaries. We begin motivating this choice by reviewing key aspects of the geometry of ReLU networks.

ReLU networks have piecewise linear boundaries that are defined by the status of each ReLU neuron in the model (Jordan et al., 2019; Hanin & Rolnick, 2019). To see this, let $u_i(\mathbf{x})$ denote the pre-activation value of the neuron $u_i$ in the network $f$ at $\mathbf{x}$. We can associate a half-space $A_i$ in the input space with the linear activation constraint $u_i(\mathbf{x}) \geq 0$ corresponding to the *activation status* of neuron $u_i$, and an *activation pattern* for a network at $\mathbf{x}$, $p(\mathbf{x})$, as the activation status of every neuron in the network. An *activation region* for a given activation pattern $p$, denoted $\mathcal{R}(p)$, is then a subspace of the network's input that yields the activations in $p$; geometrically, this is a polytope given by the convex intersection of all the half-spaces described by $p$, with facets corresponding to each neuron's activation constraint.

Note that for points in a given activation region $\mathcal{R}(p)$, the network $f$ can be expressed as a linear function, i.e. $\forall \mathbf{x} \in \mathcal{R}(p).f(\mathbf{x}) = \mathbf{w}_p^\top \mathbf{x} + b_p$ where $\mathbf{w}_p = \partial f(\mathbf{x})/\partial \mathbf{x}$ (Jordan et al., 2019; Hanin & Rolnick, 2019). Decision boundaries are thus piecewise-linear constraints, $f(\mathbf{x}) \geq 0$ for binary classifiers, or $f_i(\mathbf{x}) \geq f_j(\mathbf{x})$ between classes $i$ and $j$ for a categorical classifier, with linear pieces corresponding to the activation region of $\mathbf{x}$. This leads to the following: *(1)* if a decision boundary crosses $\mathcal{R}(p)$, then $\mathbf{w}_p$ will be orthogonal to that

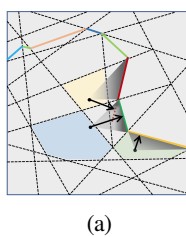
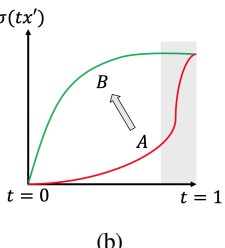

(a)                                                                (b)

Figure 2: (a) A geometric view of the input space in a ReLU network. Dashed lines correspond to activation constraints while the colorful solid lines are piece-wise linear decision boundaries. Taking gradient of the model's output with respect to the input returns a vector that is orthogonal to a nearby boundary (points in the blue and green regions) or an extension of a nearby boundary (the point in the yellow region). (b) Curves of the model's sigmoid output $\sigma(t\mathbf{x}')$ (y-axis) against interpolation parameter $t$ (x-axis).

boundary, and *(2)* if a decision boundary does not cross the region $\mathcal{R}(p)$, then $\mathbf{w}_p$ is orthogonal to an *extension* of a nearby boundary (Fromherz et al., 2021; Wang et al., 2021). In either case, the gradient with respect to the input captures information about some nearby decision boundary. Figure 2a summarizes this visually.

This analysis motivates the introduction of *distributional influence* (Definition 4), which aggregates the gradients of the model at points in a given *distribution of interest* (DoI) around $\mathbf{x}$.

**Definition 4** (Distributional Influence (Leino et al., 2018))**.** *Given an input $\mathbf{x}$, a network $f : \mathbb{R}^d \to \mathbb{R}^m$, a class of interest $c$, and a distribution of interest $\mathcal{D}_\mathbf{x}$ which describes a reference neighborhood around $\mathbf{x}$, define the distributional influence as $\chi^c_{\mathcal{D}_\mathbf{x}}(\mathbf{x}) \overset{\text{def}}{=} \mathbb{E}_{\mathbf{x}' \sim \mathcal{D}_\mathbf{x}}[\partial f_c(\mathbf{x}')/\partial \mathbf{x}']$. We write $S(\mathcal{D}_\mathbf{x})$ to represent the support of $\mathcal{D}_\mathbf{x}$. When $m = 1$, we write $\chi_{\mathcal{D}_\mathbf{x}}(\mathbf{x}) \overset{\text{def}}{=} \mathbb{E}_{\mathbf{x}' \sim \mathcal{D}_\mathbf{x}}[\partial f(\mathbf{x}')/\partial \mathbf{x}']$.*

In Leino et al. (2018), distributional influence is used to attribute the importance of a model's input and internal features on observed outcomes. Following the connection between gradients and decision boundaries in ReLU networks, we leverage it to capture useful information about nearby decision boundaries as detailed in Section 4.2.

## 4.2 Consistency and Continuity

Characterizing the precise effect changes such as random initialization have on the outcome of training is challenging. We approach this by modeling the differences that arise from small changes such as a *fine-tuning* of the original model, where the top layer of the model is re-trained and the parameters of rest of the layers are frozen.

We now introduce Theorem 2, which bounds the change on distributional influence when the model is fine-tuned at its top layer in terms of the model's Lipschitz continuity on the support of $\mathcal{D}_\mathbf{x}$. This suggests that finding a high-confidence counterfactual example in a neighborhood with a lower Lipschitz constant may lead to lower invalidation after fine-tuning, given the relationship between nearby boundaries and influence described in the previous section.

**Theorem 2.** *Let $f(\mathbf{x}) \overset{\text{def}}{=} \mathbf{w}^\top \cdot h(\mathbf{x}) + b$ be a ReLU network with a single logit output (i.e., a binary classifier), where $h(\mathbf{x})$ is the output of the penultimate layer, and denote $\sigma_\mathbf{w} = \sigma(f(\mathbf{x}))$ as the sigmoid output of the model at $\mathbf{x}$. Let $\mathcal{W} \overset{\text{def}}{=} \{\mathbf{w}' : ||\mathbf{w} - \mathbf{w}'|| \leq \Delta\}$ and $\chi_{\mathcal{D}_\mathbf{x}}(\mathbf{x}; \mathbf{w})$ be the distributional influence of $f$ when weights $\mathbf{w}$ are used at the top layer. If $h$ is $K$-Lipschitz in the support $S(\mathcal{D}_\mathbf{x})$, the following inequality holds:*

$$\forall \mathbf{w}' \in \mathcal{W}, \quad ||\chi_{\mathcal{D}_\mathbf{x}}(\mathbf{x}; \mathbf{w}) - \chi_{\mathcal{D}_\mathbf{x}}(\mathbf{x}; \mathbf{w}')|| \leq K \sqrt{[d\sigma(\mathbf{x}; \mathbf{w})||\mathbf{w}|| + C_1]^2 + C_2}$$

*where $C_1$ and $C_2$ are constants and $d\sigma(\mathbf{x}; \mathbf{w}) \overset{\text{def}}{=} \partial \sigma_\mathbf{w} / \partial f$.*

**Observations.**    Theorem 2 characterizes the extent to which a model's local decision boundaries, by proxy of influence, may change as a result of fine-tuning. This intuitively relates to the likelihood of a counterfactual's invalidation, as a point near a decision boundary undergoing a large shift is more likely to experience a change in prediction than one near a stable portion of the boundary. As the two key ingredients in Theorem 2 are the local Lipschitz constant and the model's confidence at $\mathbf{x}$, this suggests that searching for high-confidence points in neighborhoods with small Lipschitz constants will yield more

consistent counterfactuals. While Theorem 2 does not provide a direct bound on invalidation, and is limited to changes only at the network's top layer, we characterize the effectiveness of this heuristic in more general settings empirically in Section 5 after showing how to efficiently operationalize it in Section 4.3.

## 4.3 FINDING CONSISTENT COUNTERFACTUALS

The results from the previous section suggest that counterfactuals with higher sigmoid output and lower Lipschitz Constants of the penultimate layer's output with respect to the DoI $\mathcal{D}_\mathbf{x}$ will be more consistent across related models. *Stable Neighbor Search* (SNS) leverages this intuition to find consistent counterfactuals by searching for those with a low Lipschitz constant and confident counterfactual. We can find such points with the objective in Equation 1, which assumes a given counterfactual point $\mathbf{x}$.

$$\mathbf{x}_c = \arg\max_{\mathbf{x}' \in B(\mathbf{x},\delta)} [\sigma(\mathbf{x}') - K_{S'}] \quad \text{such that } F(\mathbf{x}_c;\theta) = F(\mathbf{x};\theta) \tag{1}$$

In Eq. 1 above and throughout this section, we assume that $F$ is a binary classifier with a single-logit output $f$, and sigmoid output $\sigma(f(\mathbf{x}))$. When $f$ is clear from the context, we directly write $\sigma(\mathbf{x})$. The results are readily extended to multi-logit outputs by optimizing over the maximal logit at $\mathbf{x}$. $K_{S'}$ is the Lipschitz Constant of the model's sigmoid output over the support $S(\mathcal{D}_{\mathbf{x}'})$. We relax the Lipschitz constant $K$ of the penultimate output in the Theorem 2 to the Lipschitz constant of the entire network, as in practice any parameter in the network, and not just the top layer, may change.

Leveraging a well-known relationship between the dual norm of the gradient and a function's Lipschitz constant (Paulavičius & Žilinskas, 2006), we can rephrase this objective as shown in Equation 2. Note that we assume $\ell_2$ norms throughout, so the dual remains $\ell_2$.

$$\mathbf{x}_c = \arg\max_{\mathbf{x}' \in B(\mathbf{x},\delta)} \left[\sigma(\mathbf{x}') - \max_{\hat{\mathbf{x}} \in S(\mathcal{D}_{\mathbf{x}'})} \|\frac{\partial \sigma(\hat{\mathbf{x}})}{\partial \hat{\mathbf{x}}}\|\right] \quad \text{such that } F(\mathbf{x}_c;\theta) = F(\mathbf{x};\theta) \tag{2}$$

**Choice of DoI.** The choice of DoI determines the neighborhood of points from which we gain an understanding of the local decision boundary (Wang et al., 2021). In this paper, following prior work, we choose $\mathcal{D}$ as $\text{Uniform}(\mathbf{0} \to \mathbf{x})$, a uniform distribution over a linear path between a zero vector and the current input (Sundararajan et al., 2017). That is, the set of points in $\mathcal{D}$ is $S(\mathcal{D}) \overset{\text{def}}{=} \{t\mathbf{x}, t \in [0,1]\}$. Equation 3 below updates the objective accordingly.

$$\mathbf{x}_c = \arg\max_{\mathbf{x}' \in B(\mathbf{x},\delta)} \left[\sigma(\mathbf{x}') - \max_{t \in [0,1]} \|\frac{\partial \sigma(t\mathbf{x}')}{\partial (t\mathbf{x}')}\|\right] \quad \text{such that } F(\mathbf{x}_c;\theta) = F(\mathbf{x};\theta) \tag{3}$$

While Equation (3) provides an objective that uses only primitives that are readily available in most neural network frameworks, solving the inner objective using gradient descent requires second-order derivatives of the network, which is computationally prohibitive. In the following, we discuss a sequence of relaxations to Eq. (3) that provides resource-efficient objective function.

**Avoiding vacuous second-order derivatives.** There exists a lower-bound of the term $\max_{t \in [0,1]} \|\partial\sigma(t\mathbf{x}')/\partial(t\mathbf{x}')\|$ by utilizing the following Proposition 1, which allows us to relax Eq. 3 by maximizing a differentiable lower-bound of the gradient norm rather than the gradient norm itself.

**Proposition 1.** *Let $q$ be a differentiable, real-valued function in $\mathbb{R}^d$ and $S$ be the support set of $\text{Uniform}(\mathbf{0} \to \mathbf{x})$. Then for $\mathbf{x}' \in S$, $\|\partial q(\mathbf{x}')/\partial \mathbf{x}'\| \geq \|\mathbf{x}\|^{-1} |\partial q(r\mathbf{x}')/\partial r|_{r=1}|$.*

Noting that the constant factor $\|\mathbf{x}\|$ is irrelevant to the desired optimization problem, Equation 4 below updates the objective by fitting $\sigma$ into the place of $q$ in Proposition 1. The absolute-value operator is omitted because the derivative of the sigmoid function is always non-negative.

$$\mathbf{x}_c = \arg\max_{\mathbf{x}' \in B(\mathbf{x},\delta)} \left[\sigma(\mathbf{x}') - \max_{t \in [0,1]} \frac{\partial \sigma(t\mathbf{x}')}{\partial t}\right] \quad \text{such that } F(\mathbf{x}_c;\theta) = F(\mathbf{x};\theta) \tag{4}$$

The second term in Equation 4, $-\max_{t \in [0,1]} \partial\sigma(t\mathbf{x}')/\partial t$, is interpreted by plotting the output score $\sigma(t\mathbf{x}')$ against the interpolation variable $t$ as shown in Fig. 2b. This term encourages finding a counterfactual point $\mathbf{x}_c$ where the outputs of the model for points between the zero vector ($t = 0$) and itself ($t = 1$) form a smooth and flattened curve B in Fig. 2b. Therefore, by incorporating the graph interpretation

of $-\max_{t \in [0,1]} \partial \sigma(t\mathbf{x}')/\partial t$ to find an solution of $\mathbf{x}_c$ that corresponds to curve B, we can instead try to increase the area under the curve of $\sigma(t\mathbf{x}')$ against $t$, which simplifies our objective function with replacing the inner-derivative with an integral shown in Equation 5.

$$\mathbf{x}_c = \arg \max_{\mathbf{x}' \in B(\mathbf{x}, \delta)} \left[ \sigma(\mathbf{x}') + \int_0^1 \sigma(t\mathbf{x}')dt \right] \quad \text{such that } F(\mathbf{x}_c; \theta) = F(\mathbf{x}; \theta) \tag{5}$$

One observation of the objective defined by Equation 5 is that the first term $\sigma(\mathbf{x}')$ is redundant, as differentiating the second integral term already provides useful gradient information to increase $\sigma(\mathbf{x}')$. Equation 5 thus yields our approach, *Stable Neighbor Search*.

**Definition 5** (Stable Neighbor Search (SNS)). *Given a starting counterfactual* $\mathbf{x}$ *for a network* $F(\mathbf{x})$, *its* stable neighbor $\mathbf{x}_c$ *of radius* $\epsilon$ *is the solution to the following objective:*

$$\arg \max_{\mathbf{x}' \in B(\mathbf{x}, \delta)} \int_0^1 \sigma(t\mathbf{x}')dt$$

We implement the integral in Definition 5 as a summation over a grid of points of a specified resolution, which controls the quality of the approximation. The complexity of SNS is linear in the number of points in this grid.

## 5 EVALUATION

In this section, we evaluate the extent of invalidation across five different counterfactual generation methods, including Stable Neighbor Search, over models trained with two sources of randomness in setup: *1)* initial weights, and *2)* leave-one-out differences in training data. Our results show that Stable Neighbor Search consistently generates counterfactuals with lower invalidation rates than all other methods, in many cases eliminating invalidation altogether on tested points. Additionally, despite not explicitly minimizing cost, SNS counterfactuals manage to maintain low cost relative to other methods that aim to minimize invalidation.

### 5.1 SETUP

**Data.** Our experiments encompass several tabular classification datasets from the UCI database including: German Credit, Taiwanese Credit-Default, Seizure, and Cardiotocography (CTG). We also include FICO HELOC (FICO, 2018a) and Warfarin Dosing (Consortium, 2009). All datasets have two classes except Warfarin, where we assume that the most favorable outcome (class 0) is the desired counterfactual for the other classes, and vice versa. Further details of these datasets are included in Appendix B.1.

**Baselines.** We compare SNS with the following baselines in terms of the invalidation rate. Further details about how we implement and configure these techniques are found in Appendix B.3. **Min-Cost** $\ell_1/\ell_2$ (Wachter et al., 2018): we implement this by setting the appropriate parameters for the elastic-net loss (Chen et al., 2018) in ART (Nicolae et al., 2018). **Min-Cost** $\epsilon$**-PGD** (Wachter et al., 2018): We perform Projected Gradient Descent (PGD) for an increasing sequence of $\epsilon$ until a counterfactual is found. **Pawelczyk et al.** (Pawelczyk et al., 2020b): This method attempts to find counterfactual examples *on the data manifold*, that are therefore more resistant to invalidation, by searching the latent space of a variational autoencoder, rather than the input space. **Looveren et al.** (Van Looveren & Klaise, 2019): This method minimizes an elastic loss combined with a term that encourages finding examples on the data manifold.

We note that PGD was originally proposed in the context of adversarial adversarial examples (Szegedy et al., 2013). As has been noted in prior work, the problem of finding adversarial examples is mathematically identical to that of finding counterfactual examples (Freiesleben, 2020; Browne & Swift, 2020; Sokol & Flach, 2019; Wachter et al., 2018). While solution sparsity is sometimes noted as a differentiator between the two, we note that techniques from both areas of research can be used with various $\ell_p$ metrics. We measure cost in terms of both $\ell_1$ and $\ell_2$ norms, providing $\ell_2$ in the main body and $\ell_1$ in Appendix B.6.

**Implementation of SNS.** SNS begins with a given counterfactual example as mentioned in Def. 5, which we generate with Min. $\ell_1/\ell_2$ and Min. $\epsilon$ PGD. We use the sum of 10 points to approximate the integral.

*Invalidation Rate*

| Method | German Credit | | Seizure | | CTG | | Warfarin | | HELOC | | Taiwanese Credit | |
|---|---|---|---|---|---|---|---|---|---|---|---|---|
| | LOO | RS | LOO | RS | LOO | RS | LOO | RS | LOO | RS | LOO | RS |
| Min. $\ell_1$ | 0.41 | 0.56 | - | - | 0.07 | 0.29 | 0.44 | 0.35 | 0.30 | 0.43 | 0.30 | 0.78 |
| +SNS | 0.00 | 0.07 | - | - | **0.00** | 0.01 | **0.00** | **0.00** | **0.00** | **0.00** | 0.00 | 0.04 |
| Min. $\ell_2$ | 0.36 | 0.56 | 0.64 | 0.77 | 0.48 | 0.49 | 0.35 | 0.3 | 0.55 | 0.61 | 0.27 | 0.72 |
| +SNS | 0.00 | **0.06** | **0.02** | 0.13 | **0.00** | **0.00** | **0.00** | **0.00** | **0.00** | **0.00** | 0.00 | **0.04** |
| Min. $\epsilon$ PGD | 0.28 | 0.61 | 0.94 | 0.94 | 0.04 | 0.09 | 0.10 | 0.12 | 0.04 | 0.11 | 0.04 | 0.24 |
| +SNS | **0.00** | 0.12 | 0.04 | 0.16 | **0.00** | **0.00** | 0.01 | 0.02 | **0.00** | **0.00** | **0.00** | 0.11 |
| Looveren et al. | 0.25 | 0.40 | 0.48 | 0.54 | 0.11 | 0.18 | 0.26 | 0.25 | 0.25 | 0.34 | 0.29 | 0.53 |
| Pawelczyk et al. | 0.20 | 0.35 | 0.16 | **0.11** | 0.00 | 0.06 | 0.02 | 0.01 | 0.05 | 0.15 | 0.02 | 0.20 |

*Counterfactual Cost ($\ell_2$)*

| Method | German Credit | Seizure | CTG | Warfarin | HELOC | Taiwanese Credit |
|---|---|---|---|---|---|---|
| Min. $\ell_1$ | 1.33 | - | 0.17 | 0.50 | 0.24 | 1.56 |
| Min. $\ell_2$ | 4.49 | 8.23 | 0.06 | 0.54 | 0.11 | 2.65 |
| Looveren et al. | 5.37 | 8.40 | 0.11 | 1.03 | 0.45 | 2.82 |
| Min. $\epsilon$ PGD | 1.02 | 1.36 | 0.08 | 0.31 | 0.32 | 0.75 |
| Min.$\ell_1$ + SNS | 3.40 | - | 0.25 | 0.80 | 1.71 | 3.50 |
| Min.$\ell_2$ + SNS | 6.23 | 9.60 | **0.21** | 0.90 | 1.71 | 4.68 |
| PGD + SNS | **3.03** | **3.60** | 0.22 | **0.50** | 1.79 | **2.78** |
| Pawelczyk et al. | 7.15 | 13.66 | 1.07 | 2.62 | **1.35** | 4.24 |

*IV - Cost Correlation*

| $R^2$ | 0.05 | 0.06 | 0.02 | 0.01 | 0.17 | 0.05 |
|---|---|---|---|---|---|---|

Table 1: The consistency of counterfactuals measured by invalidation rates. The average $\ell_2$ cost of different methods are also included. Results are aggregated over 100 networks for each experiment (RS and LOO). Lower invalidation rates and cost are more desirable. For $\ell_2$ cost, the best results are highlighted among three methods (separated by a line) with lower invalidation rates. If a method has significantly low success rate in generating counterfactual examples, we report '-'. In the last line, we present the $R^2$ correlation coefficient from a linear regression predicting invalidation percentage from cost. Small values indicate weak correlation.

**Retraining Controls.** We prepare different models for the same dataset using Tensorflow 2.3.0 and all computations are done using a Titan RTX accelerator on a machine with 64 gigabytes of memory. We control the random seeds used by both numpy and Tensorflow, and enable deterministic GPU operations in Tensorflow (tensorflow-determinism Python package). We evaluate the invalidation rate of counterfactual examples under changes in retraining stemming from the following two sources (see Appendix B.4 for more details on our training setup). **Leave-One-Out (LOO):** We select a random point (without replacement) to remove from the training data. Network parameters are initialized with the same values across runs. **Random Seed (RS):** Network parameters are initialized by incrementing the random seed across runs, while other hyperparameters and the data remain fixed.

We note that these sources of variation do not encompass the full set of sources that we are relevant to counterfactual invalidation, such as fine-tuning and changes in architecture or other hyperparameters. However, they are straightforward to control, produce very similar models that nonetheless tend to invalidate counterfactuals, and they are not dependent on any deployment or data-specific considerations in the way that fine-tuning changes would be. While we hope that our results are indicative of what might be observed across other sources, exploring invalidation in more depth in particular applications is important future work.

**Metrics.** To benchmark the consistency of counterfactuals generated by different algorithms, we compute the mean invalidation rate (Def. 2) over the validation split of each dataset. To calculate the extent of correlation between cost and invalidation, as discussed in Section 3, we perform a linear regression (`scipy.linregress`) between the costs for each valid counterfactual, across all five methods, with its invalidation rate across both LOO and RS differences. Table 1 reports the resulting $R^2$ for each dataset.

**Methodology.** For each dataset, we train a "base" model and compute counterfactual examples using the five methods for each point in the validation split. For each set of experiments (LOO or RS), we train 100 additional models, and compute the invalidation rate between the base model and the 100 variants. The results are shown in Table 1.

## 5.2 RESULTS

Looking at the invalidation results in Table 1, the most salient trend is apparent in the low invalidation rates of SNS compared to the other methods. SNS achieves the lowest invalidation rate across all datasets in both LOO and RS experiments, except for on the Seizure dataset with RS variations, where there is a two-point difference in the invalidation rate. SNS generates counterfactuals with *no* invalidation on CTG, Warfarin, and Heloc, and no invalidation over LOO differences on German Credit and Taiwanese Credit.

Notably, this is down from invalidation rates as high as $61\%$ from other methods on Heloc, and $\approx 10-50\%$ on others. On Seizure, which had IV rates as high as $94\%$ from other methods, SNS achieves just $2\%$ (LOO) invalidation. The closest competitor is the method of Pawelczyk et al. (2020b), which achieves zero invalidation in one case (CTG under LOO), but at significantly greater cost –in five out of six cases, SNS produced less-costly counterfactuals, and in nearly every case the margin between the two is greater than $2\times$.

As discussed in Section 3, while increasing cost is not a reliable way to generate stable counterfactuals for deep models, our results do show that stable counterfactuals tend to be more costly. The data suggests that greater-than-minimal cost appears to be necessary for stability. While SNS counterfactuals are much less costly than those generated by Pawelczyk et. al, they are consistently more costly than other methods that aim minimize cost without other constraints. To investigate the relationship between counterfactual cost and invalidation more closely, we report the $R^2$ coefficient of determination of a linear regression between the cost of each valid counterfactual generated and its invalidation rate in Table 1. Recall that a $R^2$ ranges from zero to one, with scores closer to zero indicating no linear relationship. Notably, Table 1 shows that the correlation between cost and invalidation is quite weak: the *maximum $R^2$* over all datasets is $0.17$ (Heloc), while most of the other datasets report coefficients that are much smaller–at or below $0.05$.

## 6 RELATED WORK

Counterfactual examples enjoy popularity in the research literature (Sokol & Flach, 2019; Wachter et al., 2018; Keane & Smyth, 2020; Dandl et al., 2020; Van Looveren & Klaise, 2019; Mahajan et al., 2019; Yang et al., 2020; Verma et al., 2020; Pawelczyk et al., 2020a; Dhurandhar et al., 2018; Guidotti et al., 2018), especially in the wake of legislation increasing legal requirements on explanations of machine learning models (Kaminski, 2019; GDPR, 2016). However, recent work has pointed to problems with counterfactual examples that could occur during deployment (Laugel et al., 2019; Pawelczyk et al., 2020b; Barocas et al., 2020; Rawal et al., 2021). For example, Barocas & Selbst (2016) point to the tension between the usefulness of a counterfactual and the ability to keep the explained model private. Previous work investigating the problem of invalidation, has pointed to cost as a heuristic for evaluating counterfactual invalidation at generation time (Pawelczyk et al., 2020b; Rawal et al., 2021). We demonstrate that cost is not a reliable metric for predicting invalidation in *deep* models, and show how the Lipschitz constant and confidence of a model around a counterfactual can be a more faithful guide to finding stable counterfactual examples.

While in this work, we address the problem of multiplicitious deep models producing varying outputs on *counterfactual examples*, recent work has shown that there are large differences in model prediction behavior on *any* input across small changes to the model (Black & Fredrikson, 2021; Marx et al., 2019; D'Amour et al., 2020). Instability has also been shown to be a problem for gradient-based explanations, although this is largely studied in an adversarial context (Dombrowski et al., 2019; Ghorbani et al., 2019; Heo et al., 2019).

Within the related field of adversarial examples, there is a recent interest in *adversarial transferability* (Dong et al., 2018; Ilyas et al., 2019; Xie et al., 2019), where adversarial attacks are induced to transfer between models. In general, adversarial transferability concerns transferring attacks between extremely different models—e.g., trained on disjoint training sets. Meanwhile, in this work, we decrease counterfactual invalidation between very *similar* models, in order to preserve recourse and explanation consistency. Interestingly, Goodfellow et al. (2014) suggest that transferability of adversarial examples is due to local linearity in deep networks. This supports our motivation: we find stable counterfactuals in more Lipschitz regions of the model, i.e. where it behaves (approximately) linearly. We note, however, that as linearity does not imply Lipschitzness, this insight does not provide a clear path to generating stable counterfactuals. We look forward to exploring the potential overlap between these two areas as future work.

## 7 CONCLUSION

In this paper, we characterize the consistency of counterfactual examples in deep models, and demonstrate that counterfactual cost and consistency are not strongly correlated. To mitigate the problem of counterfactual inconsistency, we introduce *Stable Neighbor Search*, which finds stable counterfactuals by leveraging the connection between the Lipschitz constant and confidence of the network around a counterfactual, and its consistency. At a high level, our work adds to the growing perspective in the field of explainability that creating good explanations requires good models to begin with.

## ACKNOWLEDGMENTS

This work was developed with the support of NSF grant CNS-1704845, CNS-1943016, as well as by DARPA and the Air Force Research Laboratory under agreement number FA8750-15-2-0277. The U.S. Government is authorized to reproduce and distribute reprints for Governmental purposes not withstanding any copyright notation thereon. The views, opinions, and/or findings expressed are those of the author(s) and should not be interpreted as representing the official views or policies of DARPA, the Air Force Research Lab- oratory, the National Science Foundation, or the U.S. Government.

## ETHICS

This paper demonstrates the problem of counterfactual invalidation in deep networks, and introduces a counterfactual generation method, Stable Neighbor Search (SNS), which creates counterfactual examples which yield consistent outcomes across nearby models. We note that the increased stability in counterfactual examples which SNS provides may eventually factor in to an engineer's, lawmaker's, or business' decision about what type of model to use: with the potential for more stable explanations, deep networks may seem more favorable. This, along with the ever-increasing zeal to incorporate neural networks in more applications, may lead practitioners to choose a deep model, when a simpler model may be a better fit for orthogonal reasons. However, if used wisely, we believe SNS can lead to positive impacts, by lessening the invalidation of recourse to users who desire a different model outcome.

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

## A PROOFS

### A.1 THEOREM 1 AND LEMMA 1

**Theorem 1** *Suppose that $H_1, H_2$ are orthognal decision boundaries in a piecewise-linear network $F(\mathbf{x}) = sign\{w_1^\top ReLU(W_0 \mathbf{x})\}$, and let $\mathbf{x}$ be an arbitrary point in its domain. If the projections of $\mathbf{x}$ onto the corresponding halfspace constraints of $H_1, H_2$ are on $H_1$ and $H_2$, then there exists a point $\mathbf{x}'$ such that:*

$$1)\, d(\mathbf{x}', H_2) = 0 \qquad 2)\, d(\mathbf{x}', H_2) < d(\mathbf{x}, H_2) \qquad 3)\, d(\mathbf{x}, H_1) \leq d(\mathbf{x}', H_1)$$

*where $d(\mathbf{x}, H_*)$ denotes the distance between $\mathbf{x}$ and the nearest point on a boundary $H_*$.*

*Proof.* Let $u(\mathbf{x})_i = W_0 \mathbf{x}$ be the pre-activation of the neuron $i$-th output in the hidden layer. The status of the neuron therefore will have the following two status: ON if $u(\mathbf{x})_i > 0$ and OFF otherwise. When a neuron is ON, the post-activation is identical to the pre-activation. Therefore, we can represent the ReLU function as a linear function of all neurons' activation status. Formally, the logit output of the network $F$ can be written as

$$f(\mathbf{x}) = w_1^\top \Lambda W_0 \mathbf{x} \tag{6}$$

where $\Lambda$ is a diagonal matrix $diag([\lambda_0, \lambda_1, ..., \lambda_n])$ such that $\lambda_i = \mathbb{I}(u(\mathbf{x})_i > 0)$. The network is a linear function within a neighborhood if all points in such a neighborhood have the same activation matrix $\Lambda$. For any two decision boundaries $H_1$ and $H_2$, the normal vectors of these decision boundaries can be written as $\mathbf{n}_1^\top = w_1^\top \Lambda_1 W_0$ and $\mathbf{n}_2^\top = w_1^\top \Lambda_2 W_0$, respectively, where $\Lambda_1$ and $\Lambda_2$ are determined by the activation status of internal neurons.

For an input $\mathbf{x}$, if the projections of $\mathbf{x}$ onto the corresponding halfspace constraints of $H_1, H_2$ are on $H_1$ and $H_2$, then the distance $d(\mathbf{x}, H_1)$ and $d(\mathbf{x}, H_2)$ are given by projections as follows:

$$d(\mathbf{x}, H_1) = \frac{|\mathbf{n}_1^\top \mathbf{x}|}{||\mathbf{n}_1||_2}, \quad d(\mathbf{x}, H_2) = \frac{|\mathbf{n}_2^\top \mathbf{x}|}{||\mathbf{n}_2||_2} \tag{7}$$

W.L.O.G. we assume $F(\mathbf{x}) = 1$ and $\mathbf{n}_1$ and $\mathbf{n}_2$ point towards $\mathbf{x}$. Let a point $\mathbf{y}$ defined as

$$\mathbf{y} = \mathbf{y}' - \frac{|\mathbf{n}_2^\top \mathbf{y}'| \mathbf{n}_2}{||\mathbf{n}_2||_2^2} \tag{8}$$

$$\mathbf{y}' = \mathbf{x} + \eta \frac{\mathbf{n}_1}{||\mathbf{n}_1||_2} \tag{9}$$

where $\eta$ is tiny positive scalar such that $F(\mathbf{y}) = F(\mathbf{x}) = 1$. We firstly show that $d(\mathbf{y}, H_2) = 0$ as follows:

$$d(\mathbf{y}, H_2) = \frac{|\mathbf{n}_2^\top \mathbf{y}|}{||\mathbf{n}_2||_2} \tag{10}$$

$$= \frac{|\mathbf{n}_2^\top (\mathbf{y}' - \frac{|\mathbf{n}_2^\top \mathbf{y}'| \mathbf{n}_2}{||\mathbf{n}_2||_2^2})|}{||\mathbf{n}_2||_2} \tag{11}$$

$$= \frac{|\mathbf{n}_2^\top \mathbf{y}' - |\mathbf{n}_2^\top \mathbf{y}'||}{||\mathbf{n}_2||_2} \tag{12}$$

$$= \frac{|\mathbf{n}_2^\top \mathbf{y}' - \mathbf{n}_2^\top \mathbf{y}'|}{||\mathbf{n}_2||_2} \quad (\eta \text{ is tiny so } \mathbf{n}_2 \text{ points to } \mathbf{y}') \tag{13}$$

$$= 0 \tag{14}$$

We secondly show that $d(\mathbf{y}, H_1) \geq d(\mathbf{x}, H_1)$ as follows:

$$d(\mathbf{y}, H_1) = \frac{|\mathbf{n}_1^\top \mathbf{y}|}{||\mathbf{n}_1||_2} \tag{15}$$

$$= \frac{|\mathbf{n}_1^\top (\mathbf{y}' - \frac{|\mathbf{n}_2^\top \mathbf{y}'|\mathbf{n}_2}{||\mathbf{n}_2||_2^2})|}{||\mathbf{n}_1||_2} \tag{16}$$

$$= \frac{|\mathbf{n}_1^\top (\mathbf{x} + \eta\frac{\mathbf{n}_1}{||\mathbf{n}_1||_2} - \frac{|\mathbf{n}_2^\top (\mathbf{x} + \eta\frac{\mathbf{n}_1}{||\mathbf{n}_1||_2})|\mathbf{n}_2}{||\mathbf{n}_2||_2^2})|}{||\mathbf{n}_1||_2} \tag{17}$$

$$= \frac{|\mathbf{n}_1^\top \mathbf{x} + \eta||\mathbf{n}_1||_2 - \mathbf{n}_1^\top \mathbf{n}_2 \frac{|\mathbf{n}_2^\top (\mathbf{x} + \eta\frac{\mathbf{n}_1}{||\mathbf{n}_1||_2})|}{||\mathbf{n}_2||_2^2})|}{||\mathbf{n}_1||_2} \tag{18}$$

$$= \frac{|\mathbf{n}_1^\top \mathbf{x} + \eta||\mathbf{n}_1||_2|}{||\mathbf{n}_1||_2} \quad (H_1 \text{ and } H_2 \text{ are orthogonal}) \tag{19}$$

$$\geq \frac{|\mathbf{n}_1^\top \mathbf{x}|}{||\mathbf{n}_1||_2} = d(\mathbf{x}, H_1) \tag{20}$$

The proof of Theorem 1 is complete. $\square$

**Lemma 1** *Let $H_1, H_2, F$ and $\mathbf{x}$ be defined as in Theorem 1. If the projections of $\mathbf{x}$ onto the corresponding halfspace constraints of $H_1, H_2$ are on $H_1$ and $H_2$, but there* does not *exist a point $\mathbf{x}'$ satisfying* (2) *and* (3) *from Theorem 1, then $H_1 = H_2$.*

Note if we remove the assumption that $H_1$ and $H_2$ are orthogonal, we will show that Theorem 1 will hold by condition. Let $m(\mathbf{x}) = |\mathbf{n}_1^\top \mathbf{x} + \eta||\mathbf{n}_1||_2 - \mathbf{n}_1^\top \mathbf{n}_2 \frac{|\mathbf{n}_2^\top (\mathbf{x} + \eta\frac{\mathbf{n}_1}{||\mathbf{n}_1||_2})|}{||\mathbf{n}_2||_2^2})|$. Assume the angle between the normal vectors of $H_1$ and $H_2$ is $\theta$ such that $\mathbf{n}_1^\top \mathbf{n}_2 = ||\mathbf{n}_1||_2 ||\mathbf{n}_2||_2 \cos\theta$.

$$m(\mathbf{x}) = |\mathbf{n}_1^\top \mathbf{x} + \eta||\mathbf{n}_1||_2 - \mathbf{n}_1^\top \mathbf{n}_2 \frac{|\mathbf{n}_2^\top (\mathbf{x} + \eta\frac{\mathbf{n}_1}{||\mathbf{n}_1||_2})|}{||\mathbf{n}_2||_2^2})| \tag{21}$$

$$= |\mathbf{n}_1^\top \mathbf{x} + \eta(||\mathbf{n}_1||_2 - \frac{\mathbf{n}_1^\top \mathbf{n}_2 \cdot \mathbf{n}_2^\top \mathbf{n}_1}{||\mathbf{n}_2||_2^2 ||\mathbf{n}_1||_2}) - \frac{\mathbf{n}_1^\top \mathbf{n}_2 \cdot \mathbf{n}_2^\top \mathbf{x}}{||\mathbf{n}_2||_2^2}| \tag{22}$$

$$= |\mathbf{n}_1^\top \mathbf{x} + \eta(1 - \cos^2\theta)||\mathbf{n}_1||_2 - \mathbf{n}_1 \mathbf{x} \cos\theta| \tag{23}$$

Since $d(\mathbf{y}, H_1) \propto m(\mathbf{x})$ and $d(\mathbf{x}, H_1) \propto |\mathbf{n}_1^\top \mathbf{x}|$ and they share they same denominator $||\mathbf{n}_1||_2$. In order to have $m(\mathbf{x}) > |\mathbf{n}_1^\top \mathbf{x}|$, we just need $\eta(1 - \cos^2\theta)||\mathbf{n}_1||_2 - \mathbf{n}_1 \mathbf{x} \cos\theta > 0$, which means we need to find a $\eta$ such that $\eta(1 - \cos^2\theta)||\mathbf{n}_1||_2 > \mathbf{n}_1 \mathbf{x} \cos\theta$. Moving terms around we have the following inequality:

$$\eta > \frac{\mathbf{n}_1 \mathbf{x} \cos\theta}{(1 - \cos^2\theta)||\mathbf{n}_1||_2} = \frac{||\mathbf{x}||_2}{\frac{1}{\cos\theta} - \cos\theta} \tag{24}$$

The RHS goes to 0 when $\theta \to \frac{\pi}{2}$, which corresponds to the situation of Theorem 1. When $\theta \to 0$ ($H_1 = H_2$), RHS goes to $\infty$, which means we cannot find a point $\mathbf{y}$ satisfying the Theorem 1, which completes the proof of Lemma 1.

## A.2 THEOREM 2 AND PROPOSITION 1

**Theorem 2** *Let $f(\mathbf{x}) \overset{\text{def}}{=} \mathbf{w}^\top \cdot h(\mathbf{x}) + b$ be a ReLU network with a single logit output (i.e., a binary classifier), where $h(\mathbf{x})$ is the output of the penultimate layer, and denote $\sigma_{\mathbf{w}} = \sigma(f(\mathbf{x}))$ as the sigmoid output of the model at $\mathbf{x}$. Let $\mathcal{W} \overset{\text{def}}{=} \{\mathbf{w}' : ||\mathbf{w} - \mathbf{w}'|| \leq \Delta\}$, $\chi_{\mathcal{D}_\mathbf{x}}(\mathbf{x})$ be the distributional influence of $f$ when weights $\mathbf{w}$ are used at the top layer, and $\chi'_{\mathcal{D}_\mathbf{x}}(\mathbf{x})$ be the distributional influence of $f$ when weights $\mathbf{w}'$ are used at the top layer. If $h$ is $K$-Lipschitz in the support $S(\mathcal{D}_\mathbf{x})$, the following inequality holds:*

$$\forall \mathbf{w}' \in \mathcal{W}, \quad ||\chi_{\mathcal{D}_\mathbf{x}}(\mathbf{x}; \mathbf{w}) - \chi_{\mathcal{D}_\mathbf{x}}(\mathbf{x}; \mathbf{w}')|| \leq K\sqrt{[d\sigma(\mathbf{x}; \mathbf{w})||\mathbf{w}|| + C_1]^2 + C_2}$$

*where $C_1$ and $C_2$ are constants and $d\sigma(\mathbf{x}; \mathbf{w}) \overset{\text{def}}{=} \partial\sigma_{\mathbf{w}}/\partial f$.*

*Proof.* Consider a ReLU network as $g(h(\mathbf{x}))$. We first write out the expression of $h(x)$:

$$h(x) = \phi_{N-1}(W_{N-1}(\cdots\phi_1(W_1 x + b_1)) + b_{N-2}) \tag{25}$$

where $W_i, b_i$ are the parameters for the $i$-th layer and $\phi_i(\cdot)$ is the corresponding ReLU activation. By the definition of the distributional influence,

$$\chi_{\mathcal{D}}(\mathbf{x}) = \mathbb{E}_{\mathbf{z}\sim\mathcal{D}(\mathbf{x})} \frac{\partial\sigma(g(h(\mathbf{z};\mathbf{w})))}{\partial\mathbf{z}} \tag{26}$$

$$= \mathbb{E}_{\mathbf{z}\sim\mathcal{D}(\mathbf{x})} \frac{\sigma(g)}{\partial g} \frac{\partial g(h;\mathbf{w})}{\partial h} \frac{\partial h(\mathbf{z})}{\partial\mathbf{z}} \tag{27}$$

$$= \mathbb{E}_{\mathbf{z}\sim\mathcal{D}(\mathbf{x})} \left[ \sigma(\mathbf{z};\mathbf{w})(1-\sigma(\mathbf{z};\mathbf{w}))\mathbf{w} \prod_{i=1}^{N-1} (W_i\Lambda_i(\mathbf{z}))^\top \right] \tag{28}$$

$$\tag{29}$$

where $W_i$ is the weight of the layer $l_i$ if $l_i$ is a dense layer or the equivalent weight matrix of a convolutional layer and $\Lambda_i(\mathbf{z})$ is an diagonal matrix with each diagonal entry being 1 if the neuron is activated or 0 other wise when evaluated at the point $\mathbf{z}$.

$$||\chi_{\mathcal{D}}(\mathbf{x}) - \chi'_{\mathcal{D}}(\mathbf{x})|| \tag{30}$$

$$= ||\mathbb{E}_{\mathbf{z}\sim\mathcal{D}(\mathbf{x})} \left[ \sigma(\mathbf{z};\mathbf{w})(1-\sigma(\mathbf{z};\mathbf{w}))\mathbf{w} \prod_{i=1}^{N-1} (W_i\Lambda_i(\mathbf{z}))^\top \right] \tag{31}$$

$$- \mathbb{E}_{\mathbf{z}\sim\mathcal{D}(\mathbf{x})} \left[ \sigma(\mathbf{z};\mathbf{w}')(1-\sigma(\mathbf{z};\mathbf{w}'))\mathbf{w}' \prod_{i=1}^{N-1} (W_i\Lambda_i(\mathbf{z}))^\top \right] || \tag{32}$$

$$= ||\mathbb{E}_{\mathbf{z}\sim\mathcal{D}(\mathbf{x})} \left[ (\sigma(\mathbf{z};\mathbf{w})(1-\sigma(\mathbf{z};\mathbf{w}))\mathbf{w} - \sigma(\mathbf{z};\mathbf{w}')(1-\sigma(\mathbf{z};\mathbf{w}'))\mathbf{w}') \prod_{i=1}^{N-1} (W_i\Lambda_i(\mathbf{z}))^\top \right] || \tag{33}$$

$$\leq \mathbb{E}_{\mathbf{z}\sim\mathcal{D}(\mathbf{x})} \left[ ||(\sigma(\mathbf{z};\mathbf{w})(1-\sigma(\mathbf{z};\mathbf{w}))\mathbf{w} - \sigma(\mathbf{z};\mathbf{w}')(1-\sigma(\mathbf{z};\mathbf{w}'))\mathbf{w}') \prod_{i=1}^{N-1} (W_i\Lambda_i(\mathbf{z}))^\top || \right] \tag{34}$$

(Jensen's Inequality from (33) to (34))

$$\leq \mathbb{E}_{\mathbf{z}\sim\mathcal{D}(\mathbf{x})} \left[ ||(\sigma(\mathbf{z};\mathbf{w})(1-\sigma(\mathbf{z};\mathbf{w}))\mathbf{w} - \sigma(\mathbf{z};\mathbf{w}')(1-\sigma(\mathbf{z};\mathbf{w}'))\mathbf{w}')|| \cdot || \prod_{i=1}^{N-1} (W_i\Lambda_i(\mathbf{z}))^\top || \right] \tag{35}$$

(By the definition of matrix operator norm from (34) to (35))

To simplify the expression, we denote

$$\mathbf{a} = \sigma(\mathbf{z};\mathbf{w})(1-\sigma(\mathbf{z};\mathbf{w}))\mathbf{w} - \sigma(\mathbf{z};\mathbf{w}')(1-\sigma(\mathbf{z};\mathbf{w}'))\mathbf{w}' \tag{36}$$

$$\mathbf{B} = \prod_{i=1}^{N-1} (W_i\Lambda_i(\mathbf{z}))^\top \tag{37}$$

and now Equation (35) now becomes

$$||\chi_{\mathcal{D}}(\mathbf{x}) - \chi'_{\mathcal{D}}(\mathbf{x})|| \leq \mathbb{E}_{\mathbf{z}\sim\mathcal{D}(\mathbf{x})}[||\mathbf{a}|| \cdot ||\mathbf{B}||] \tag{38}$$

with Cauchy-Shwartz inequality, we find that

$$\mathbb{E}_{\mathbf{z}\sim\mathcal{D}(\mathbf{x})}[||\mathbf{a}|| \cdot ||\mathbf{B}||] \leq \sqrt{\mathbb{E}_{\mathbf{z}\sim\mathcal{D}(\mathbf{x})}||\mathbf{a}||^2} \cdot \sqrt{\mathbb{E}_{\mathbf{z}\sim\mathcal{D}(\mathbf{x})}||\mathbf{B}||^2} \tag{39}$$

Now we will show that these two terms $\sqrt{\mathbb{E}_{\mathbf{z}\sim\mathcal{D}(\mathbf{x})}||\mathbf{a}||^2}$ and $\sqrt{\mathbb{E}_{\mathbf{z}\sim\mathcal{D}(\mathbf{x})}||\mathbf{B}||^2}$ are bounded.

(1) Bound for the term $\sqrt{\mathbb{E}_{\mathbf{z}\sim\mathcal{D}(\mathbf{x})}||\mathbf{a}||^2}$

Consider the relation between the expectation and the variance of random variables

$$\mathbb{E}X^2 = (\mathbb{E}X)^2 + Var(X) \tag{40}$$

which implies that

$$\mathbb{E}_{\mathbf{z}\sim\mathcal{D}(\mathbf{x})}||\mathbf{a}||^2 = (\mathbb{E}_{\mathbf{z}\sim\mathcal{D}(\mathbf{x})}||\mathbf{a}||)^2 + Var(||\mathbf{a}||) \tag{41}$$

We simplify the notation by defining

$$d\sigma(\mathbf{z};\mathbf{w}') \stackrel{\text{def}}{=} \sigma(\mathbf{z};\mathbf{w}')(1-\sigma(\mathbf{z};\mathbf{w}'))$$

and we denote $d\sigma(\mathbf{z};\mathbf{w}) = d\sigma(\mathbf{x};\mathbf{w}) + \delta(\mathbf{z};\mathbf{w})$ and $d\sigma(\mathbf{z};\mathbf{w}') = d\sigma(\mathbf{x};\mathbf{w}') + \delta(\mathbf{z};\mathbf{w}')$. Note that $\delta \leq \frac{1}{4}$ because $d\sigma \in [0, \frac{1}{4}]$. Therefore, the $\mathbb{E}_{\mathbf{z}\sim\mathcal{D}(\mathbf{x})}||\mathbf{a}||$ can be simplified as

$$\mathbb{E}_{\mathbf{z}\sim\mathcal{D}(\mathbf{x})}||\mathbf{a}|| = \mathbb{E}_{\mathbf{z}\sim\mathcal{D}(\mathbf{x})}||d\sigma(\mathbf{x};\mathbf{w})\mathbf{w} - d\sigma(\mathbf{x};\mathbf{w}')\mathbf{w}' + \delta(\mathbf{z};\mathbf{w})\mathbf{w} - \delta(\mathbf{z};\mathbf{w}')\mathbf{w}'|| \tag{42}$$

$$\leq ||d\sigma(\mathbf{x};\mathbf{w})\mathbf{w} - d\sigma(\mathbf{x};\mathbf{w}')\mathbf{w}'|| + \mathbb{E}_{\mathbf{z}\sim\mathcal{D}}||\delta(\mathbf{z};\mathbf{w})\mathbf{w} - \delta(\mathbf{z};\mathbf{w}')\mathbf{w}'|| \quad \text{(Triangle Inequality)} \tag{43}$$

$$\leq ||d\sigma(\mathbf{x};\mathbf{w})\mathbf{w} - d\sigma(\mathbf{x};\mathbf{w}')\mathbf{w}'|| + \mathbb{E}_{\mathbf{z}\sim\mathcal{D}}||\delta(\mathbf{z};\mathbf{w})\mathbf{w}|| + \mathbb{E}_{\mathbf{z}\sim\mathcal{D}}||\delta(\mathbf{z};\mathbf{w}')\mathbf{w}'|| \tag{44}$$

$$\leq ||d\sigma(\mathbf{x};\mathbf{w})\mathbf{w} - d\sigma(\mathbf{x};\mathbf{w}')\mathbf{w}'|| + \mathbb{E}_{\mathbf{z}\sim\mathcal{D}}|\delta(\mathbf{z};\mathbf{w})|||\mathbf{w}|| + \mathbb{E}_{\mathbf{z}\sim\mathcal{D}}|\delta(\mathbf{z};\mathbf{w}')|||\mathbf{w}'|| \tag{45}$$

$$\leq ||d\sigma(\mathbf{x};\mathbf{w})\mathbf{w} - d\sigma(\mathbf{x};\mathbf{w}')\mathbf{w}'|| + \frac{1}{4}(||\mathbf{w}|| + ||\mathbf{w}'||) \qquad (\delta \leq \frac{1}{4}) \tag{46}$$

$$\leq ||d\sigma(\mathbf{x};\mathbf{w})\mathbf{w} - d\sigma(\mathbf{x};\mathbf{w}')\mathbf{w}'|| + \frac{1}{2}(||\mathbf{w}|| + \frac{1}{2}\Delta) \qquad (||w - w'|| \leq \Delta) \tag{47}$$

$$\leq ||d\sigma(\mathbf{x};\mathbf{w})\mathbf{w}|| + ||d\sigma(\mathbf{x};\mathbf{w}')\mathbf{w}'|| + \frac{1}{2}(||\mathbf{w}|| + \frac{1}{2}\Delta) \quad \text{(Triangle Inequality)} \tag{48}$$

$$\leq d\sigma(\mathbf{x};\mathbf{w})||\mathbf{w}|| + \frac{1}{4}(||\mathbf{w}|| + \Delta) + \frac{1}{2}(||\mathbf{w}|| + \frac{1}{2}\Delta) \quad (d\sigma \leq \frac{1}{4}) \tag{49}$$

$$\leq d\sigma(\mathbf{x};\mathbf{w})||\mathbf{w}|| + \frac{3}{4}||\mathbf{w}|| + \frac{1}{2}\Delta \tag{50}$$

To summarize, we find that

$$\mathbb{E}_{\mathbf{z}\sim\mathcal{D}(\mathbf{x})}||\mathbf{a}|| \leq d\sigma(\mathbf{x};\mathbf{w})||\mathbf{w}|| + \frac{3}{4}||\mathbf{w}|| + \frac{1}{2}\Delta \tag{51}$$

Since both sides of the inequalities are non-negative scalars, we see that

$$(\mathbb{E}_{\mathbf{z}\sim\mathcal{D}(\mathbf{x})}||\mathbf{a}||)^2 \leq \left[ d\sigma(\mathbf{x};\mathbf{w})||\mathbf{w}|| + \frac{3}{4}||\mathbf{w}|| + \frac{1}{2}\Delta \right]^2 \tag{52}$$

The derivation of the variance term $Var(||\mathbf{a}||)$ may not have a simple analytical form to show that it is bounded, but it is easy to find an upper bound of $||\mathbf{a}||$

$$||\mathbf{a}|| = ||d\sigma(\mathbf{x};\mathbf{w})\mathbf{w} - d\sigma(\mathbf{x};\mathbf{w}')\mathbf{w}' + \delta(\mathbf{z};\mathbf{w})\mathbf{w} - \delta(\mathbf{z};\mathbf{w}')\mathbf{w}'|| \quad \text{(from Equation (43))} \tag{53}$$

$$\leq ||d\sigma(\mathbf{x};\mathbf{w})\mathbf{w}|| + ||d\sigma(\mathbf{x};\mathbf{w}')\mathbf{w}'|| + ||\delta(\mathbf{z};\mathbf{w})\mathbf{w}|| + ||\delta(\mathbf{z};\mathbf{w}')\mathbf{w}'|| \quad \text{(Triangle Inequality)} \tag{54}$$

$$\leq \frac{1}{4}||\mathbf{w}|| + \frac{1}{4}(||\mathbf{w}|| + \Delta) + ||\mathbf{w}|| + (||\mathbf{w}|| + \Delta) \quad (d\sigma \leq \frac{1}{4}), (||w - w'|| \leq \Delta) \tag{55}$$

$$\leq \frac{5}{2}||\mathbf{w}|| + \frac{5}{4}\Delta \tag{56}$$

which implies that $||\mathbf{a}|| \in [0, \frac{5}{2}||\mathbf{w}|| + \frac{5}{4}\Delta]$. With Popoviciu's inequality, the variance $Var(||\mathbf{a}||)$ must be bounded such that

$$Var(||\mathbf{a}||) \leq \frac{1}{4}\left[ \frac{5}{2}||\mathbf{w}|| + \frac{5}{4}\Delta \right]^2 \tag{57}$$

Now so far we have derived the upper-bounds for $(\mathbb{E}_{\mathbf{z}\sim\mathcal{D}(\mathbf{x})}||\mathbf{a}||)^2$ and $Var(||\mathbf{a}||)$; put together, we show that

$$\sqrt{\mathbb{E}_{\mathbf{z}\sim\mathcal{D}(\mathbf{x})}||\mathbf{a}||^2}\leq\sqrt{\left[d\sigma(\mathbf{x};\mathbf{w})||\mathbf{w}||+C_1\right]^2+C_2} \tag{58}$$

where

$$C_1=\frac{3}{4}||\mathbf{w}||+\frac{1}{2}\Delta \tag{59}$$

$$C_2=\frac{1}{4}\left[\frac{5}{2}||\mathbf{w}||+\frac{5}{4}\Delta\right]^2 \tag{60}$$

(2) Bound for the term $\sqrt{\mathbb{E}_{\mathbf{z}\sim\mathcal{D}(\mathbf{x})}||\mathbf{B}||^2}$

$||\mathbf{B}||$ is the operator norm, namely the spectral norm for the matrix $\mathbf{B}$. As we assume $h(\mathbf{x})$ is $K$-Lipschitz, we know that

$$\mathbb{E}_{\mathbf{z}\sim\mathcal{D}(\mathbf{x})}||\mathbf{B}||=\mathbb{E}_{\mathbf{z}\sim\mathcal{D}(\mathbf{x})}||\left[\prod_{i=1}^{N-1}(W_i\Lambda_i(\mathbf{z}))^\top\right]||\leq\sup_{\mathbf{z}\sim\mathcal{D}(\mathbf{x})}||\left[\prod_{i=1}^{N-1}(W_i\Lambda_i(\mathbf{z}))^\top\right]||=K \tag{61}$$

$$\tag{62}$$

and $\forall\mathbf{z}\sim\mathcal{D}(\mathbf{x})$

$$||\mathbf{B}||\leq\sup_{\mathbf{z}\sim\mathcal{D}(\mathbf{x})}||\left[\prod_{i=1}^{N-1}(W_i\Lambda_i(\mathbf{z}))^\top\right]||=K \tag{63}$$

$$\tag{64}$$

Therefore,

$$||\mathbf{B}||^2\leq K^2 \tag{65}$$

$$\mathbb{E}_{\mathbf{z}\sim\mathcal{D}(\mathbf{x})}||\mathbf{B}||^2\leq K^2 \tag{66}$$

which implies $\sqrt{\mathbb{E}_{\mathbf{z}\sim\mathcal{D}(\mathbf{x})}||\mathbf{B}||^2}\leq K$. To put together, we finish the proof and show that

$$||\chi_\mathcal{D}(\mathbf{x})-\chi'_\mathcal{D}(\mathbf{x})||\leq\sqrt{\mathbb{E}_{\mathbf{z}\sim\mathcal{D}(\mathbf{x})}||\mathbf{a}||^2}\cdot\sqrt{\mathbb{E}_{\mathbf{z}\sim\mathcal{D}(\mathbf{x})}||\mathbf{B}||^2}\leq K\sqrt{\left[d\sigma(\mathbf{x};\mathbf{w})||\mathbf{w}||+C_1\right]^2+C_2} \tag{67}$$

$\square$

## A.3 PROPOSITION 1

**Proposition 1** *Let $q$ be a differentiable, real-valued function in $\mathbb{R}^d$ and $S$ be the support set of Uniform$(\mathbf{0}\to\mathbf{x})$. $\forall\mathbf{x}'\in S$,*

$$||\frac{\partial q(\mathbf{x}')}{\partial\mathbf{x}'}||\geq||\mathbf{x}||^{-1}|\frac{\partial q(r\mathbf{x}')}{\partial r}|_{r=1}|$$

*Proof.* First, we show that $\forall\mathbf{x}'\in S$

$$|\frac{\partial q(\mathbf{x}')}{\partial\mathbf{x}'}^\top\cdot\mathbf{x}'|\leq||\frac{\partial q(\mathbf{x}')}{\partial\mathbf{x}'}||\cdot||\mathbf{x}'|| \quad \text{(Cauchy–Schwarz)} \tag{68}$$

By the construction of $\mathbf{x}'$ we know $||\mathbf{x}'|| \leq ||\mathbf{x}||$; therefore,

$$|\frac{\partial q(\mathbf{x}')}{\partial \mathbf{x}'}^\top \cdot \mathbf{x}'| \leq ||\frac{\partial q(\mathbf{x}')}{\partial \mathbf{x}'}|| \cdot ||\mathbf{x}|| \tag{69}$$

$$||\frac{\partial q(\mathbf{x}')}{\partial \mathbf{x}'}|| \geq ||\mathbf{x}||^{-1} |\frac{\partial q(\mathbf{x}')}{\partial \mathbf{x}'}^\top \cdot \mathbf{x}'| \tag{70}$$

Now consider a function $p(r;\mathbf{x}') = r\mathbf{x}'$. Then we show a trick of chain rule.

$$\frac{\partial q(p)}{\partial r} = \frac{\partial q(p)}{p}^\top \cdot \frac{\partial p(r;\mathbf{x}')}{\partial r} = \frac{\partial q(p)}{p}^\top \cdot \mathbf{x}' \tag{71}$$

Replacing the notation $p$ with $\mathbf{x}'$ in $\frac{\partial q(\mathbf{x}')}{\partial \mathbf{x}'}^\top$ does not change the computation of taking the Jacobian of $q$'s output with respect to the input; therefore, we show that

$$\frac{\partial q(\mathbf{x}')}{\partial \mathbf{x}'}^\top \cdot \mathbf{x}' = \frac{\partial q(p)}{p}^\top \big|_{r=1} \cdot \mathbf{x}' = \frac{\partial q(p)}{\partial r}\big|_{r=1} = \frac{\partial q(r\mathbf{x}')}{\partial r}\big|_{r=1} \tag{72}$$

We therefore complete the proof of Proposition 1 by showing

$$||\frac{\partial q(\mathbf{x}')}{\partial \mathbf{x}'}|| \geq ||\mathbf{x}||^{-1} |\frac{\partial q(r\mathbf{x}')}{\partial r}\big|_{r=1}| \tag{73}$$

$\square$

## B  EXPERIMENT DETAILS

### B.1  META INFORMATION FOR DATASETS

The German Credit Dua & Karra Taniskidou (2017) and Taiwanese Credit Dua & Karra Taniskidou (2017) data sets consist of individuals financial data, with a binary response indicating their creditworthiness. For the German Credit Dua & Karra Taniskidou (2017) dataset, there are 1000 points, and 20 attributes. We one-hot encode the data to get 61 features, and standardize the data to zero mean and unit variance using SKLearn Standard scaler. We partitioned the data intro a training set of 700 and a test set of 200. The Taiwanese Credit Dua & Karra Taniskidou (2017) dataset has 30,000 instances with 24 attributes. We one-hot encode the data to get 32 features and normalize the data to be between zero and one. We partitioned the data intro a training set of 22500 and a test set of 7500.

The HELOC dataset FICO (2018a) contains anonymized information about the Home Equity Line of Credit applications by homeowners in the US, with a binary response indicating whether or not the applicant has even been more than 90 days delinquent for a payment. The dataset consists of 10459 rows and 23 features, some of which we one-hot encode to get a dataset of 10459 rows and 40 features. We normalize all features to be between zero and one, and create a train split of 7,844 and a validation split of 2,615.

The Seizure Dua & Karra Taniskidou (2017) dataset comprises time-series EEG recordings for 500 individuals, with a binary response indicating the occurrence of a seizure. This is represented as 11500 rows with 178 features each. We split this into 7,950 train points and 3,550 test points. We standardize the numeric features to zero mean and unit variance.

The CTG Dua & Karra Taniskidou (2017) dataset comprises of 2126 fetal cardiotocograms processed and labeled by expert obstetricians into three classes of fetuses, healthy, suspect, and pathological. We have turned this into a binary response between healthy and other classes. We split the data into 1,700 train points and a validation split of 425. There are 21 features for each instance, which we normalize to be between zero and one.

The Warfain dataset is collected by the International Warfarin Pharmacogenetics Consortium Consortium (2009) about patients who were prescribed warfarin. We removed rows with missing values, 4819 patients remained in the dataset. The inputs to the model are demographic (age, height, weight, race), medical (use of amiodarone, use of enzyme inducer), and genetic (VKORC1, CYP2C9) attributes. Age, height, and weight are real-valued and were scaled to zero mean and unit variance. The medical attributes take binary

values, and the remaining attributes were one-hot encoded. The output is the weekly dose of warfarin in milligrams, which we encode as "low", "medium", or "high", following Consortium (2009).

The UCI datasets are under an MIT license, and Warfarin datasets are under a Creative Commons License. Dua & Karra Taniskidou (2017); Consortium (2009). The license for the FICO HELOC dataset is available at the dataset challenge website, and allows use for research purposes FICO (2018b).

## B.2 Hyper-parameters and Model Architectures

The German Credit and Seizure models have three hidden layers, of size 128, 64, and 16. Models on the Taiwanese dataset have two hidden layers of 32 and 16, and models on the HELOC dataset have two deep layers with sizes 100 and 32. The Warfarin models have one hidden layer of 100. The CTG models have three layers, of sizes 100, 32, and 16. German Credit, Adult, Seizure, Taiwanese, CTG and Warfarin models are trained for 100 epochs; HELOC models are trained for 50 epochs. German Credit models are trained with a batch size of 32; Adult, Seizure, and Warfarin models with batch sizes of 128; Taiwanese Credit models with batch sizes of 512, and CTG models with a batch size of 16. All models are trained with keras' Adam optimizer with the default parameters.

## B.3 Implementation of Baseline Methods

We describe the parameters specific to each baseline method here. Common choices of hyper-parameters are shown in Table 2.

**Min-Cost $\ell_1/\ell_2$ Wachter et al. (2018)** We implement this by setting $\beta = 1.0$ for $\ell_1$ (or $\beta = 0.0$ for $\ell_2$) and `confidence`=0.5 for the elastic-net loss Chen et al. (2018) in ART Nicolae et al. (2018).

**Min-$\epsilon$ PGD Madry et al. (2018):** For a given $\epsilon$, we use 10 interpolations between 0 and the current $\epsilon$ as the norm bound in each PGD attack. The step size is set to $2 * \epsilon_c /$ `max_steps` where $\epsilon_c$ is the norm bound used. The maximum allowed norm bound is the median of the $\ell_2$ norm of data points in the training set.

**Pawelczyk et al. Pawelczyk et al. (2020b):** We train an AutoEncoder (AE) instead of a Variational AutoEncoder (VAE) to estimate the data manifold. Given that VAE jointly estimate the mean and the standard deviation of the latent distribution, it creates non-deterministic latent representation for the same input. In the contact with Pawelczyk et al., we are informed that we can only use the mean as the latent representation for an input; therefore, by taking out the standard deviation from a VAE, we instead train a AE that produces deterministic latent representation for each input. When searching for the latent representation of a counterfactual, we use random search as proposed by Pawelczyk et al. Pawelczyk et al. (2020b): we randomly sample 1280 points around the latet representation of an input within a norm bound of 1.0 in the latent space. When generating random points, we use a fixed random seed 2021. If there are multiple counterfactuals, we return the one that is closest to the input. For all datasets, we use the following architecture for the hidden layers: 1024-128-32-128-1024.

**Looveren et al. Van Looveren & Klaise (2019):** We use the public implementation of this method[2]. We use k-d trees with $k = 20$ to estimate the data manifold as the curre implementation only supports an AE where the input features must be between 0 and 1, while our dataset are not normalized into this range. The rest of the hyper-parameters are default values from the implementation: `theta`=100, `max_iterations`=100. This implementation only supports for non-eager mode so we turn off the eager execution in TF2 by running `tf.compat.v1.disable_eager_execution()` for this baseline.

**SNS** : We run SNS for 200 steps for all datasets and project the counterfactual back to a $\ell_2$ ball. The size of the ball is set to be 0.8 multiplied by the largest size of the ball used for the baseline Min-$\epsilon$ PGD. For Max $\ell_1/\ell_2$ without a norm bound, we use the norm bound from Min-$\epsilon$ PGD. Similarly, the step size is set to $2 * 0.8 * \epsilon / 200$.

---

[2]https://docs.seldon.io/projects/alibi/en/stable/methods/CFProto.html

*Hyper-parameters and Success Rate*

| **Min $\ell_1$** | German Credit | Seizure | CTG | Warfarin | HELOC | Taiwanese Credit |
|---|---|---|---|---|---|---|
| $\epsilon$ | - | - | - | - | - | - |
| `step size` | 0.05 | 0.05 | 0.05 | 0.5 | 0.01 | 0.05 |
| `success rate` | 0.35 | 0.14 | 1.00 | 1.00 | 1.00 | 1.00 |

| **Min $\ell_2$** | German Credit | Seizure | CTG | Warfarin | HELOC | Taiwanese Credit |
|---|---|---|---|---|---|---|
| $\epsilon$ | - | - | - | - | - | - |
| `step size` | 0.01 | 0.01 | 0.01 | 0.01 | 0.01 | 0.01 |
| `success rate` | 0.84 | 0.71 | 1.00 | 1.00 | 1.00 | 1.00 |

| **Min $\epsilon$ PGD** | German Credit | Seizure | CTG | Warfarin | HELOC | Taiwanese Credit |
|---|---|---|---|---|---|---|
| Max. $\epsilon$ | 3.00 | 3.00 | 0.20 | 0.50 | 2.10 | 5.00 |
| `step size` | `adp.` | `adp.` | `adp.` | `adp.` | `adp.` | `adp.` |
| `success rate` | 0.90 | 0.86 | 0.51 | 0.85 | 1.00 | 1.00 |

| **Looveren et al.** | German Credit | Seizure | CTG | Warfarin | HELOC | Taiwanese Credit |
|---|---|---|---|---|---|---|
| $\epsilon$ | - | - | - | - | - | - |
| `step size` | - | - | - | - | - | - |
| `success rate` | 1.0 | 1.0 | 1.0 | 1.0 | 1.0 | 1.0 |

| **Pawelczyk et al.** | German Credit | Seizure | CTG | Warfarin | HELOC | Taiwanese Credit |
|---|---|---|---|---|---|---|
| $\epsilon$ | 0.3 | 1.0 | 1.0 | 1.0 | 1.0 | 1.0 |
| `step size` | - | - | - | - | - | - |
| `success rate` | 0.38 | 1.00 | 0.87 | 0.72 | 0.76 | 0.14 |

Table 2: Hyper-parameters and Success Rate for each baseline methods. `adp.` denotes that the step size for each iteration is $2*\epsilon/$`max_steps`.

### B.4 Details of Retraining

We evaluate counterfactual invalidation over models with one-point differences in their training set, or different random initialization. For each dataset, we train a base model $F(\theta)$ with a specified random seed to determine initialization, and a specified train-validation split. We use this to generate all counterfactuals. We then train 100 models with one-point differences in the training set from a base model, as well as 100 models trained with different random initialization parameters. To do this, we randomly derive: a training set $S$, a set $O \subseteq S$ of size 100 that consists of points drawn randomly from test data (i.e. with which to create 100 different training sets with one point removed, $S^{(\setminus i)}$), and a test set. Then, For each $z_i \in O$, we train $F(\theta)'$ on $S^{(\setminus i)}$ by removing $z_i$ from $S$. To train the 100 models with different initialization parameters, we simply change the numpy random seed directly before initializing a model.

### B.5 Full results of IV with Standard Deviations

The full results of invalidation rates with standard deviations are shown in Table 3.

### B.6 $\ell_1$ and $\ell_2$ results

The full results of $\ell_1$ and $\ell_2$ costs with standard deviations are shown in Table 4.

*Invalidation Rate (LOO)*

| Method | German Credit | Seizure | CTG | Warfarin | HELOC | Taiwanese Credit |
|---|---|---|---|---|---|---|
| Min. $\ell_1$ | 0.41±0.04 | - | 0.07±0.09 | 0.44±0.02 | 0.30±0.03 | 0.30±0.03 |
| +SNS | 0.00±0.00 | - | 0.00±0.00 | 0.00±0.00 | 0.00±0.00 | 0.00±0.00 |
| Min. $\ell_2$ | 0.36±0.05 | 0.64±0.06 | 0.48±0.17 | 0.35±0.02 | 0.55±0.05 | 0.27±0.05 |
| +SNS | 0.00±0.00 | 0.02±0.02 | 0.00±0.00 | 0.00±0.00 | 0.00±0.00 | 0.00±0.00 |
| Min. $\epsilon$ PGD | 0.28±0.03 | 0.94±0.01 | 0.04±0.03 | 0.10±0.01 | 0.04±.01 | 0.04±0.00 |
| +SNS | 0.00±0.00 | 0.04±0.02 | 0.00±0.00 | 0.01±0.00 | 0.00±0.00 | 0.00±0.00 |
| Looveren et al. | 0.25±0.03 | 0.48±0.04 | 0.11±0.08 | 0.26±0.02 | 0.25±0.03 | 0.29±0.06 |
| Pawelczyk et al. | 0.20±0.13 | 0.16±0.14 | 0.00±0.00 | 0.02±0.00 | 0.05±0.06 | 0.02±0.01 |

*Invalidation Rate (RS)*

| Method | German Credit | Seizure | CTG | Warfarin | HELOC | Taiwanese Credit |
|---|---|---|---|---|---|---|
| Min. $\ell_1$ | 0.56±0.05 | - | 0.29±0.09 | 0.35±0.08 | 0.43±0.07 | 0.78±0.06 |
| +SNS | 0.07±0.02 | - | 0.01±0.00 | 0.00±0.00 | 0.00±0.00 | 0.04±0.02 |
| Min. $\ell_2$ | 0.56±0.06 | 0.77±0.12 | 0.49±0.15 | 0.30±0.05 | 0.61±0.07 | 0.72±0.07 |
| +SNS | 0.06±0.04 | 0.13±0.08 | 0.00±0.00 | 0.00±0.00 | 0.00±0.00 | 0.04±0.04 |
| Min. $\epsilon$ PGD | 0.61±0.04 | 0.94±0.12 | 0.09±0.04 | 0.12±0.03 | 0.11±0.02 | 0.24±0.07 |
| +SNS | 0.12±0.03 | 0.16±0.08 | 0.00±0.00 | 0.02±0.01 | 0.00±0.00 | 0.11±0.05 |
| Looveren et al. | 0.40±0.03 | 0.54±0.05 | 0.18±0.08 | 0.25±0.02 | 0.34±0.05 | 0.53±0.06 |
| Pawelczyk et al. | 0.35±0.16 | 0.11±0.17 | 0.06±0.04 | 0.01±0.00 | 0.15±0.21 | 0.20±0.09 |

Table 3: Invalidation Rates with standard deviations for each datasets and each re-training situations. Results are aggregated over 100 models.

*Counterfactual Cost ($\ell_2$)*

| Method | German Credit | Seizure | CTG | Warfarin | HELOC | Taiwanese Credit |
|---|---|---|---|---|---|---|
| Min. $\ell_1$ | 1.33±1.07 | - | 0.17±0.12 | 0.50±0.33 | 0.24±0.18 | 1.56±0.94 |
| Min. $\ell_2$ | 4.49±1.90 | 8.23±2.27 | 0.06±0.04 | 0.54±0.57 | 0.11±0.08 | 2.65±1.08 |
| Looveren et al. | 5.37±2.53 | 8.40±6.96 | 0.11±0.06 | 1.03±0.46 | 0.45±0.45 | 2.82±1.89 |
| Min. $\epsilon$ PGD | 1.02±0.57 | 1.36±0.38 | 0.08±0.03 | 0.31± 0.12 | 0.32±0.12 | 0.75±0.27 |
| Min.$\ell_1$ + SNS | 3.40±0.82 | - | 0.25±0.08 | 0.80±0.29 | 1.71±0.12 | 3.50±0.91 |
| Min.$\ell_2$ + SNS | 6.23±1.65 | 9.60±2.31 | 0.21±0.04 | 0.90±0.54 | 1.71±0.11 | 4.68±1.03 |
| PGD + SNS | 3.03±0.38 | 3.60±0.59 | 0.22±0.04 | 0.50±0.11 | 1.79±0.15 | 2.78±0.49 |
| Pawelczyk et al. | 7.15±2.12 | 13.66±7.46 | 1.07±0.20 | 2.62±0.79 | 1.35±0.93 | 4.24±2.23 |

*Counterfactual Cost ($\ell_1$)*

| Method | German Credit | Seizure | CTG | Warfarin | HELOC | Taiwanese Credit |
|---|---|---|---|---|---|---|
| Min. $\ell_1$ | 2.09±2.00 | - | 0.16±0.19 | 0.48±0.59 | 0.35±0.30 | 2.40±2.84 |
| Min. $\ell_2$ | 24.70±13.14 | 77.89±28.38 | 0.18±0.12 | 1.17±0.91 | 0.43±0.31 | 9.76±3.93 |
| Looveren et al. | 15.93±8.93 | 76.99±69.93 | 0.16±0.12 | 1.75±0.99 | 0.97±1.05 | 6.11±5.12 |
| Min. $\epsilon$ PGD | 6.31±3.56 | 14.55±4.15 | 0.30±0.12 | 1.07±0.42 | 1.45±0.54 | 3.19±1.09 |
| Min.$\ell_1$ + SNS | 13.81±2.96 | - | 0.58±0.14 | 1.61±0.52 | 7.11±0.72 | 11.04±3.17 |
| Min.$\ell_2$ + SNS | 34.38±11.54 | 96.08±27.80 | 0.57±0.10 | 2.17±0.95 | 7.18±0.73 | 16.63±3.76 |
| PGD + SNS | 14.21±2.31 | 38.55±6.36 | 0.63±0.14 | 1.41±0.36 | 7.64±0.88 | 10.19±2.54 |
| Pawelczyk et al. | 38.48±12.31 | 145.36±77.67 | 3.03±0.63 | ±6.48±2.66 | 4.51±3.52 | 12.22±7.49 |

Table 4: $\ell_1$ and $\ell_2$ costs of counterfactuals with standard deviations.

