# OpenReview forum: "Consistent Counterfactuals for Deep Models"
_ICLR.cc/2022/Conference — ICLR 2022 Poster_

### Official Review · Reviewer_76Ar · 2021-10-25

**Correctness:** 3
**Technical Novelty And Significance:** 3
**Empirical Novelty And Significance:** 3
**Recommendation:** 6
**Confidence:** 4

**Main Review:**

**Novelty/Contribution**:
To the best of my knowledge, the consistency of counterfactuals has not been extensively studied. In my opinion, the main novelty is proposing Stable Neighbor Search (SNS), which finds counterfactuals in more locally Lipschitz regions. The results for SNS are promising -- it outperforms the benchmark methods (Van Looveren and Pawelczyk).

A second contribution is the discussion of the relationship between Lipschitzness and counterfactual consistency. This contribution is less significant, in my opinion, as it is similar to discussions in the field of adversarial examples. In adversarial example literature, it is well-established that improving the Lipschitzness of a network will improve its adversarial robustness.

**Correctness**:
Generally, I believe that the central argument of the paper is correct: generating counterfactuals in areas where the network has a smaller Lipschitz constant improves the consistency of counterfactual explanations. However, I am not sure whether the proof of theorem 2 is correct. There are some steps that require further clarification.

Specifically:
-  what property are you using in the step from lines 35-36 to 37-38? $E(ab)=E(a)E(b)$ requires $a$ and $b$ to be independent, but it's not clear to me why that holds in this case.
- line 46: $d\sigma (x,w)$ can be both positive and negative, right? As such, I think there should be an absolute value sign around $d\sigma (x,w)$.
- some unclear notation: $g$ is not defined in the main text -- is it the same as $f$?


**Writing**:
Overall, very well written and easy to follow. However, I do think the authors over-state some contributions in the introduction. For example,
"we prove that counterfactual examples in a neighbourhood where the network has a small local Lipschitz constant are more consistent against changes in the training environment (Theorem 2)". Theorem 2 assumes (1) only a change in the final layer weights and (2) a relu network with a final sigmoid activation. The authors are explicit about this later on, however, I think the introduction should be written more carefully.


**Minor language/formatting suggestions**:
- Section 3: " ... counterfactuals produce inconsistent outcomes in duplicitous models up to 94% of the time"  This sentence reads as a fact, however it's shown empirically for a specific set of hyperparameters/datasets. There are likely cases when it holds for 100% of the time. As such, perhaps rephrase it as "counterfactuals often produce inconsistent outcomes. Empirically, we've found that it occurs up to 94% of the time."
- Section 4.2, sentence 1: "characterizing the precise effect such as random initialization have on" -> "characterizing the precise effect of hyperparameters, such as random initialization, have on"
- Theorem 2: space missing between w and are in the statement, and a "." instead of a "," after forall $w' \in W$. Perhaps define $h$ as a ReLU network, rather than $f$. The latter makes it easier to read (I was a bit confused about the sigmoid activation/ relu network for a moment).
- Appendix, Proof Theorem 2. Page 16: for the equation numbering, (34), (36) and (38) should not be there. This can be remedied by using * or align in latex.
- the definition of distributional influence should contain w in the main text (to be consistent with the appendix)

-----
Post-rebuttal update: I have increased my score as the authors have corrected their proof for Theorem 2.

**Summary Of The Paper:**

The paper focuses on the consistency of counterfactual explanations (i.e., the rate at which counterfactual explanations transfer across different models). The authors show that the consistency of counterfactuals does not necessarily relate to cost (i.e., the distance between input and the counterfactual). They propose an alternative theory: they investigate the relationship between local Lipschitzness and consistent counterfactuals. Based on this idea, they propose a new method (SNS) for generating counterfactual examples. Empirically, they show that SNS creates more consistent counterfactuals than van Looveren et al. and Pawelczyk et al.

**Summary Of The Review:**


Overall, I like the paper and enjoyed reading it. However, I am currently recommending a 5 as I am not convinced by the correctness of the claims in the paper. Specifically, I have some concerns about the validity of Theorem 2 (elaborated upon in my main review). If the authors are able to clear up this aspect, I'm happy to reconsider my score.

---

> ### Author Response · Authors · 2021-11-12
> **Thank you for your review**
>
> Thank you very much for your thoughtful response.
>
> Thank you for catching the mistake in the proof of our theorem. The correct inequality to apply is the special case of Holder's Inequality -- namely the Cauchy-Schwarz inequality. We invite the reviewer to check out the revision of the proof and please take the following as an overview of the major update to the proof:
>
> 1. The proof before Equation (37) remains unchanged compared to the previous version.
> 2. We apply Cauchy-Schwarz, $|\mathbb{E}XY| \leq \sqrt{\mathbb{E}X^2} \sqrt{\mathbb{E}Y^2}$ (https://en.wikipedia.org/wiki/Cauchy–Schwarz_inequality) to separate the expectations. Because we are dealing with norms, the absolute symbols can be omitted.
> 3. We derive the bound for each term from the RHS of Cauchy-Schwarz.
>
> As a result, the updated theorem motivates our method in the same way as before: it suggests that tuning the top layer will result in greater changes to influence in regions where the model is less Lipschitz and the sigmoid output is lower. Typos in notations are fixed. We appreciate the reviewer’s feedback to improve the quality of the paper.
>
> > line 46: dσ(x,w) can be both positive and negative, right? As such, I think there should be an absolute value sign around dσ(x,w)
>
> By $d\sigma (x, w)$ we mean the derivative of the sigmoid function, the final sigmoid activation with respect to its logit inputs (see Eq. 28 in the proof of Theorem 2, where we expand this derivative). As the sigmoid is monotonically increasing, its derivative will be positive.
>
> > some unclear notation: $g$ is not defined in the main text -- is it the same as $f$?
>
> Thank you for pointing this out. It is a typo. It should be $f$, the logit output of the network, instead of $g$.
>
> >However, I do think the authors over-state some contributions in the introduction.For example, "we prove that counterfactual examples in a neighbourhood where the network has a small local Lipschitz constant are more consistent against changes in the training environment (Theorem 2)". Theorem 2 assumes (1) only a change in the final layer weights and (2) a relu network with a final sigmoid activation. The authors are explicit about this later on, however, I think the introduction should be written more carefully.
>
> Thank you for pointing this out. We will revise that sentence in the introduction to the following:
> “we prove that counterfactual examples in a neighborhood where the network has a small local Lipschitz constant are more consistent across changes to the last layer of weights, which suggests that such points are more stable across small changes in the training environment (Theorem 2)".
> Please let us know if this addresses your concern.
>
> > A second contribution is the discussion of the relationship between Lipschitzness and counterfactual consistency. This contribution is less significant, in my opinion, as it is similar to discussions in the field of adversarial examples. In adversarial example literature, it is well-established that improving the Lipschitzness of a network will improve its adversarial robustness.
>
> We agree that there are interesting connections between adversarial robustness and counterfactual consistency. However, we point out that invalidation is not addressed by robustness: the former is relative to prediction differences across training runs, and the latter to differences under feature perturbations. While it is well-known that Lipschitz continuity is useful for robustness, we are not aware of work which shows that Lipschitzness helps with predictive consistency across training runs. If the reviewer is aware of such work, we are happy to discuss it in the paper. In fact, prior work [Black & Fredrikson 2021] shows that robust models may be less consistent across training runs, so in this light, our results come as somewhat of a surprise.
>
>
> > Minor formatting comments
>
> We will make the suggested updates to the paper, thank you for bringing them to our attention.
>
> We hope that this discussion has addressed some of your concerns, and are happy to answer any further questions that you might have.

---

> > ### Comment · Reviewer_76Ar · 2021-11-19
> > **Follow-up**
> >
> > Thanks for your rebuttal -- I appreciate it, and it's addressed some points in my original review.
> >
> > Thank you for the revised proof. I've taken a look and there's a couple of steps I don't follow yet. Could you clarify the following:
> > - line 48->49: what are you using in this transition?
> > - line 56->57: what are you using in this transition?
> >
> > Aside from the two points above, the revised theorem/proof looks clear/good.
> >
> > There are some minor spelling mistakes:
> > - Cautchy-Shwartz -> Cauchy-Schwarz (before line 41)
> > - bouned-> bounded (before eq 59)

---

> > > ### Author Response · Authors · 2021-11-19
> > > **Thank you for your official follow-up**
> > >
> > > Thank you for your continued follow-up.
> > >
> > > > I've taken a look and there's a couple of steps I don't follow yet. Could you clarify the following:
> > > > line 48->49: what are you using in this transition?
> > > > line 56->57: what are you using in this transition?
> > >
> > > Note that, after following some formatting suggestions, line 48-49 is now line 46-47, and line 56->57 is now line 54-55. We rely on the assumption $||w-w'|| \leq \Delta$ (stated at the beginning of the proof) so therefore  $||w|| + ||w'|| \leq ||w|| + ||w|| + \Delta = 2||w|| + \Delta$. We have made this explicit in the proof.
> > >
> > > We have addressed the rest of your minor formatting comments, as well as made the promised update to the introduction. We are happy to clarify any follow up questions.

---

> > > > ### Comment · Reviewer_76Ar · 2021-11-25
> > > > **Thank you**
> > > >
> > > > Thanks for your follow-up, and the clarifications. This discussion (and the accompanying paper edits) have addressed my original concern about the validity of the proof.

---

### Official Review · Reviewer_fWXE · 2021-10-29

**Correctness:** 2
**Technical Novelty And Significance:** 2
**Empirical Novelty And Significance:** 2
**Recommendation:** 3
**Confidence:** 3

**Main Review:**

The paper presents a nicely thought out method on providing points on a manifold that are robust to small sometimes imperceivable  changes on the training parameters.
However, the reviewer believes that the claims relating to counterfactual are based upon a not so accurate interpretation of causality.
Firstly I would invite the authors to substitute definition 1 with the appropriate counterfactual definition from the textbook by J. Pearl, 2009, chapter 7.
Moreover, I am under the impression that the problem of producing non robust counterfactuals is rooted to the fact that the methods explored by the authors do not impose any identifiability constraints that would guarantee the models learning the intended DAGs and producing accurate counterfactuals. As such I fail to understand the expectation that the models should produce correct and robust counterfactuals when there is no explicit constrain on the models to do so. The method proposed in this paper does not refer not attempt to solve any of the aforementioned points
Counterfactual stability in categorical variables , effectively a similar outcome to this paper, was already already explored in [1].

I would invite the authors to look into the field of identifiability and robustness when the appropriate causal inference methods are used and constraints enforced


[1] Counterfactual Off-Policy Evaluation with Gumbel-Max Structural Causal Models; Oberst and Sontag ; ICML 2019

**Summary Of The Paper:**

The authors propose a regularisation method that offers increased stability on counterfactual predictions when models are affected by small changes during deployment . The method is based upon a K-Lipschitz constant.

**Summary Of The Review:**

Overall the paper has merit in terms of identifying examples that are robust upon small changes in the training procedure. The claims regarding to counterfactuals, explainability and causality I believe are a stretch. As such I recommend that the paper is changed such that it is in the proper context

---

> ### Author Response · Authors · 2021-11-12
> **Thank you for your review**
>
> Thank you for your review. In this paper, we use the term “counterfactual” to refer to a point $x’$ that causes a model to predict differently than a given reference point $x$. For example, a counterfactual example for some rejected applicant under a loan application model would be a similar application from another individual who the model decided to accept. This is distinct from how the term is used widely in the causality literature, where a counterfactual is given by an intervention on a causal model that is assumed to generate data observations. Our use of the term is consistent with usage in the explainability literature [Wachter et al., 2018; Ustun et al., 2019; Sharma et al., 2019; Poyiadzi et al., 2020; Pawelczyk et al., 2020a; Van Looveren & Klaise, 2019; Mahajan et al., 2019; Verma et al., 2020; Laugel et al., 2018; Keane & Smyth, 2020.].
>
> Counterfactuals are particularly popular explanations method in business [McGrath 2018] and legal [Wachter et al., 2018] applications, due to a number of desirable properties, including compliance with EU and US laws requiring explanations in sensitive contexts (e.g. ECOA, personal finance); their ability to generate explanations for a model of arbitrary complexity; the fact that counterfactual explanations release very little information about the underlying model, and that their creation can be fully automated [Barocas et al 2019].
>
> Thus, we make no claims that the counterfactual examples in this paper reflect interventions on a causal model for the data. We realize the overlapping terminology may be confusing for those more familiar with research surrounding causal models (e.g. Oberst and Sontag 2019), and are happy to make the distinction between our focus and causal model based approaches in our introduction and related work.
>
> > “I fail to understand the expectation that the models should produce correct and robust counterfactuals”:
>
> We agree there is no a priori reason to expect that counterfactual examples should be consistent across nearby models. However, as several authors have pointed out [Barocas et al. 2019, Verma et al. 2020], the underlying assumption in almost all works done with counterfactual explanations to date assumes that the model which \emph{creates} the counterfactual explanation, and the one that eventually provides a decision about the outcome of that counterfactual, are the same. (e.g. in a loan application context, the model which gives an applicant a counterfactual explanation, and then later decides whether to give the applicant the loan when they have re-applied trying their best to match the counterfactual given to them, will be the same). However, this is not often the case, as models are updated and retrained during deployment--which produces the risk of models changing their output on counterfactual explanations during the course of deployment, (i.e. not accepting the re-application of the applicant in our example), putting recourse through counterfactual explanations in jeopardy.
>
> Creating consistent counterfactuals robust to classifier update has been identified by several in the research community as one of the main open questions surrounding their use for certain applications, such as loan granting decisions [Verma et al. 2020, Rawal et al. 2020, Barocas et al. 2019].In our work, we show, complementing prior work, that small classifier updates (e.g. changing a point in the training set) do indeed lead to large counterfactual invalidation rates (see Table 1) [Rawal et al. 2020, Pawelczyk et al. 2020]. Importantly, we find that our counterfactual generation method, SNS, alleviates counterfactual invalidation by finding counterfactual examples in regions where the model is more stable---i.e., points with high confidence in areas with low Lipschitz constants. Our results show that SNS is better than other methods attempting to achieve consistency in counterfactuals across retrainings, e.g. Pawelczyk et al 2020.
>
> We are happy to continue our discussion should there be further questions.
>
> References (beyond those already in paper)
>
> McGrath, Rory , et al. "Interpretable credit application predictions with counterfactual explanations." arXiv preprint arXiv:1811.05245 (2018).
>
> Equal Credit Opportunity Act. Bureau of Consumer Financial Protection. Nov 11 2021. URL: https://www.ftc.gov/enforcement/statutes/equal-credit-opportunity-act

---

> > ### Comment · Reviewer_fWXE · 2021-11-15
> > **Thank you for the response**
> >
> > I would like first to thank the authors for their response and the clarifications provided.
> > I  now realise that the paper approaches the problem from a non causal perspective. However given the wide use of the word counterfactual in a causal setting on sciences like medicine, epidemiology, econometrics etc, I would ask the authors to make the fact that causality is not considered explicitly clear in the paper.
> >
> > Having said this , the method, at the end of the day, asks a causal question: given the learned links between the variables in question and a point x' that is sufficiently close to x, what is the outcome of the model, all other things being equal. The model is then expected to provide a prediction that matches that of x. This begs the following questions:
> > 1) why is causality not considered here, when the nature of the research question posed is a purely causal one ?
> > 2) given that unseen confounders in form of  exogenous noise are not modelled in the method , isnt there the probability of the model actually predicting correctly the different outcome for x' ? In other words, if there is no attention payed to unseen confounders the expectation of the model giving us the same outcome might not be entirely reasonable, as the model during its retraining might have picked up some latent characteristics that guide it to the different results. The above assumption would be reasonable in the event no unseen confounders exist, then indeed we would expect the predictions to be the same.
> >
> > In other words, I still cannot understand the expectation of the models to perform causally and fairly when there is no such considerations taken.
> > I understand that this is based on the assumption noted by the authors that the underlying model is unchanged , however this assumption a) doesnt take into account unseen confounders. b ) there is no guarantee that the Neural net based model actually learns the correct underlying model (to which the assumption refers)

---

> > > ### Author Response · Authors · 2021-11-16
> > > **Thank you for your continued discussion**
> > >
> > > Thank you for following up. We are happy to continue our discussion if you have continued concerns.
> > >
> > > > I would ask the authors to make the fact that causality is not considered explicitly clear in the paper.
> > >
> > > We are happy to make this distinction clear in the introduction and related work.
> > >
> > > > Having said this , the method, at the end of the day, asks a causal question: given the learned links between the variables in question and a point x' that is sufficiently close to x, what is the outcome of the model, all other things being equal. The model is then expected to provide a prediction that matches that of x.
> > >
> > > We apologize if our explanation in our earlier response was unclear, but as we understand the sentence above, this is not a completely accurate description of what our approach, and counterfactual examples more generally, try to achieve.
> > >
> > > Given a binary classifier $h$, and a given input, $x$, a counterfactual example is a nearby input $x'$ that $h$ classifies differently, i.e., $h(x) \ne h(x')$. This would, for example, provide guidance to a person whose loan application was rejected by a model on how they may change their application to become a successful applicant *under the classification model $h$*.
> > >
> > > > there is no guarantee that the Neural net based model actually learns the correct underlying model (to which the assumption refers)
> > >
> > > We do not make the assumption that the neural network learns the correct underlying causal model, as counterfactual examples are a practical tool used for explaining model decisions and creating the possibility of recourse in situations where causal graphs and models are not considered. While we do not expect counterfactual examples to be true counterfactuals under the causal model of the data, they are still useful, as they show individuals subject to model decisions (e.g. loan applicants) how to adjust their features (i.e. applications) so that they are more likely to succeed *under the machine learning model used* next time they apply. Providing an explanation that serves this purpose is required by US Law (FCRA).
> > >
> > > However, there is a problem that has been recently discovered with counterfactuals, especially when using these *nearest* points with opposite prediction as counterfactual examples: they are unstable (change prediction outcome) across even small changes to a model’s decision boundaries, as the nearest points with opposite prediction are essentially on top of model decision boundaries. We call this counterfactual invalidation. This phenomenon complicates the possibility of recourse. Put in terms of a practical example, the problem with counterfactual invalidation is that if an individual with application $x$ is initially rejected, given some counterfactual example $x’$, and re-applies for a loan having taken the time and effort to match the counterfactual example $x’$, if the counterfactual $x’$ is invalidated, (i.e. no longer is an accepted loan under the model) they will still not get the loan---which is extremely undesirable behavior.
> > >
> > > This phenomenon of counterfactual invalidation been pointed out in the following papers, which also include some background on the field of counterfactual examples:
> > >
> > > Martin Pawelczyk, Klaus Broelemann, and Gjergji. Kasneci. On counterfactual explanations under predictive multiplicity. In Proceedings of the 36th Conference on Uncertainty in Artificial Intelligence (UAI), Proceedings of Machine Learning Research, 2020b. [Link](https://arxiv.org/abs/2010.10596)
> > >
> > > Rawal, Kaivalya, Ece Kamar, and Himabindu Lakkaraju. "Can I Still Trust You?: Understanding the Impact of Distribution Shifts on Algorithmic Recourses." arXiv preprint arXiv:2012.11788 (2020). [Link](https://arxiv.org/pdf/2012.11788.pdf)
> > >
> > > Finding a solution to the problem of counterfactual invalidation is identified as a central research question in the area in the following paper:
> > >
> > > Verma, Sahil, John Dickerson, and Keegan Hines. "Counterfactual explanations for machine learning: A review." arXiv preprint arXiv:2010.10596 (2020). [Link](https://arxiv.org/abs/2010.10596) (see Research Challenge 9).
> > > Similar views are expressed in Barocas et al. 2019 and Rawal et al. 2020. We take the fact that this problem was noted as one of the open challenges for the research area of counterfactual examples to show that there is research interest from the community in answering this question.

---

> > > > ### Author Response · Authors · 2021-11-16
> > > > **Part 2**
> > > >
> > > > In our paper, we show how to avoid the problem of invalidation by introducing Stable Neighbor Search (SNS), a method of finding counterfactuals that are still relatively low cost (i.e. such that the counterfactual example $x’$ is still close to the original input $x$) but where the chance of invalidation is very low. Thus, our counterfactual generation method, SNS, preserves the possibility of achieving recourse with counterfactual examples in deep models.
> > > >
> > > > In summary, the problem we address in this paper is an open problem in the area of counterfactual examples pointed to by several researchers. The area of counterfactual examples that we study is the subject of several papers in machine learning (some of which referenced below), and we address an important open research question within this area. We are happy to discuss this further if you have additional questions or concerns.
> > > >
> > > > Papers on counterfactual examples:
> > > >
> > > > Wachter et al., 2018; Ustun et al., 2019; Sharma et al., 2019; Poyiadzi et al., 2020; Pawelczyk et al., 2020a; Van Looveren & Klaise, 2019; Mahajan et al., 2019; Verma et al., 2020; Laugel et al., 2018; Keane & Smyth, 2020.
> > > >
> > > > If it is more convenient, here is a link to a list of counterfactual example papers with links, created by Verma et al., the authors of a survey paper on counterfactual examples (it has 42 rows) : [Link](https://docs.google.com/spreadsheets/d/15V7NOZQh4LQMkglLHtPvgcEQf_yaGNCfQHwG1zOFCz4/edit#gid=0)

---

> > > > > ### Author Response · Authors · 2021-11-19
> > > > > **Thank you and update**
> > > > >
> > > > > Thank you again for your engagement. We are writing to update that we have included a remark in the background section (Section 2) making it clear that we do not consider causality in this work, and noting the definition of counterfactual examples in the causality literature. We are happy to continue our response if there are additional concerns or questions.

---

### Official Review · Reviewer_gXiV · 2021-11-01

**Correctness:** 4
**Technical Novelty And Significance:** 3
**Empirical Novelty And Significance:** 3
**Recommendation:** 6
**Confidence:** 4

**Main Review:**

Pro:
 - Interesting and relevant topic
 - Well written (good structure and easy to follow)
 - "Sound" evaluation

Cons:
 - Relevance of counterfactuals could be made more prominent (in the introduction). Why are they so popular? etc.
 - Sec. 4.2 limited to changes in the top layer. I understand that some simplifications are necessary but this particular simplification seems very strong to me.
 - Implementation of SNS: The authors use 10 points to approximate the integral. Why exactly 10? How does the method behave for different number of points? This specific number looks like a magic number that needs further explanations.
 - I miss some comments on the computational complexity of the proposed method (SNS)

**Summary Of The Paper:**

The authors study how counterfactual explanations of a Deep Neural Network change if the initial training settings are slightly changed. They find that even small changes in the initial training settings lead to huge differences in the counterfactual explanations. They propose a method for computing more robust (i.e. consistent) counterfactual explanations.

**Summary Of The Review:**

Overall an okay paper with only minor (not too critical) issues as pointed out in the main review.

---

> ### Author Response · Authors · 2021-11-12
> **Thank you for your review**
>
> Thank you for your review. We attempt to address your questions below, and are happy to go into greater detail when necessary.
>
> > Why are counterfactuals popular?
>
> Counterfactuals are particularly popular explanations method in business  [McGrath et al. 2018] and legal [Wachter et al. 2018] applications. Importantly, counterfactual examples may offer a way to comply with regulations in the United States and Europe (i.e. Equal Credit Opportunity Act and General Data Protection Regulation), where for example explanations are required for loan decisions [Barocas et al 2019]. Secondly, they can be computed on a broad class of models at little cost, and do not reveal significant information about the underlying model. We are happy to discuss this in the paper if the reviewer believes that it would be helpful.
>
> > Implementation of SNS: The authors use 10 points to approximate the integral. Why exactly 10? How does the method behave for different number of points? This specific number looks like a magic number that needs further explanations.
>
> Approximating this path interval via a mesh grid is a common method in many explanation techniques, most notably Integrated Gradients [Sundararajan et al., 2017]. In the seminal paper introducing integrated gradients, which focuses on high-dimensional image data, they claim that between 20-300 points in the grid will approximate the integral to within 5%  [Sundararajan et al., 2017]. Our data is tabular, so the data is much lower-dimensional; and we believe thus 10 is a good approximation. Since the practice of approximating the integral through this mesh is common, we did not include a sensitivity analysis, however we are currently running this analysis and will update this review when we update the appendix with those results. We are also happy to discuss this method in the appendix.
>
> > I miss some comments on the computational complexity of the proposed method (SNS)
>
> SNS has computational complexity linear in the number of points in the interpolation of the integral. If you would find it useful to see the timing of the method, we are happy to carry out those experiments---please advise us on whether or not you would find this helpful, or if this answer does not sufficiently answer your question.
>
> We are happy to continue our discussion should you have any further questions.
>
> References (beyond those already in paper)
>
> Grath, Rory Mc, et al. "Interpretable credit application predictions with counterfactual explanations." arXiv preprint arXiv:1811.05245 (2018).
> 2018 reform of EU data protection rules. European Commission, Nov 11 2021. URL: https://ec.europa.eu/commission/sites/beta-political/files/data-protection-factsheet-changes_en.pdf.
>
> Equal Credit Opportunity Act. Bureau of Consumer Financial Protection. Nov 11 2021. URL: https://www.ftc.gov/enforcement/statutes/equal-credit-opportunity-act

---

> > ### Comment · Reviewer_gXiV · 2021-11-15
> > **Thank you for your response**
> >
> > Thank you for your clarifying response.
> >
> > >Counterfactuals are particularly popular explanations method in business [McGrath et al. 2018] and legal [Wachter et al. 2018] applications. Importantly, counterfactual examples may offer a way to comply with regulations in the United States and Europe (i.e. Equal Credit Opportunity Act and General Data Protection Regulation), where for example explanations are required for loan decisions [Barocas et al 2019]. Secondly, they can be computed on a broad class of models at little cost, and do not reveal significant information about the underlying model. We are happy to discuss this in the paper if the reviewer believes that it would be helpful.
> >
> > Yes, I think one or two additional sentences about this would improve the paper and make the importance of counterfactuals more clear to the reader.
> >
> > >SNS has computational complexity linear in the number of points in the interpolation of the integral. If you would find it useful to see the timing of the method, we are happy to carry out those experiments---please advise us on whether or not you would find this helpful, or if this answer does not sufficiently answer your question.
> >
> > No, I do not think that showing empirically timing of the the methods would provide any benefit. However, I would find it useful to mention the linear complexity in the paper -- just one sentence.

---

> > > ### Author Response · Authors · 2021-11-19
> > > **Thank you for your continued follow-up**
> > >
> > > Thank you for your follow-up.
> > > > Yes, I think one or two additional sentences about this would improve the paper and make the importance of counterfactuals more clear to the reader.
> > >
> > > > I would find it useful to mention the linear complexity in the paper -- just one sentence.
> > >
> > > We have added a sentence to the introduction stating further background on counterfactuals, and added a sentence stating the linear complexity of SNS in Section 4.

---

### Official Review · Reviewer_ZP6K · 2021-11-06

**Correctness:** 3
**Technical Novelty And Significance:** 3
**Empirical Novelty And Significance:** 3
**Recommendation:** 6
**Confidence:** 3

**Main Review:**

## Detailed summary
While previous work finds a little higher counterfactual loss leads to lower invalidation rates in linear and well calibrated models. This paper focuses on deep NNs and show that with deep models, this may not hold as higher counterfactual loss does not mean CFE is further from the decision boundary, which invalids the conclusion in linear/well calibrated models. Theorem 1 shows that there exist data points x and x', and decision boundaries H1 and H2 s.t. x' is on H2 but is further to H1 than x, due to the complex decision boundary of DNNs. Lemma 1 further verifies this by showing if x' does not exist, then H1=H2.

It proposes distributional influence (DI), which uses gradient information to explain local decision boundaries (around a sample). Distributional influence is defined as the expected gradient of the output against the input in the local neighbor region of x ($\mathcal{D}_x$).

They find a bound on the change of DI in terms of the model's Lipschitz continuity on the support of $\mathcal{D}_x$. This means finding a high confidence CFE with low Lipschitz constant would likely to have lower invalidation rate. Based on this, they propose to maximize the confidence while minimize the Lipschitz constant under the constraint that the CFE x' and x share the same predicted label. This is transformed into eq.(4) by Proposition 1, where the second term is the maximal slope in Figure 2b. This is further simplified to eq.(5).

In experiments, they use UCI datasets, FICO HELOC and Warfarin Dosing. Note that only binary labels can be used. Results show SNS can lead to much lower invalidation rate. And SNS would lead to a higher counterfactual loss in general, as expected.

## From the above, the strength includes
1. Clear theoretical justification of the proposed method, although there are some imperfections.
2. Good presentation and visualization (e.g., Fig. 2).
3. Relative comprehensive results, considering various datasets, baselines and metrics.

## Weakness includes
1. Only relatively small tabular datasets are used, which is not the most widely used case for DNNs.
2. Some details may need to be further clarified.

# Questions and detailed comments:

How to compute the integral of SNS, what would be the complexity?

There is no causal model built and therefore, why the CFE in this paper is a counterfactual. If it is, which counterfactual distribution is it sampled from?

What is a the x-axis and y-axis of Fig 2? I guess they are two input features but it is better to show them in the figure.

It is not very clear to me which terms are related to confidence in the equation of Theorem 2.

I did not fully get why small change in DI always means low invalidation. I guess this has to be shown formally.

We know DNNs tend to be over-confident for unseen x. Would this result in any issue if we aim to find CFE with high confidence?

What is the effect of relaxing the Lipschitz constant K from the penultimate output to the entire network? Would it increase or decrease, why?

It is better to clarify what is $t$ in Fig 2 since it is only defined in the later pages.

Typo in page 6. zero vector (t=1) --> (t=0)



**Summary Of The Paper:**

This paper aims to generate valid counterfactual examples (CFE) in a dynamic setting where the model can be retrained in the test time. The goal is to minimize the invalidation rate. Different from previous work, they focus on DNNs. Through a series of theoretical analysis, they propose to maximize the confidence and minimize the Lipschitz constant under the constraint that the CFE x' and x share the same predicted label.  Experiments on tabular datasets show the proposed method reduces the invalidation rate a lot.


**Summary Of The Review:**

Copied from above.

## Strength
1. Clear theoretical justification of the proposed method, although there are some imperfections.
2. Good presentation and visualization (e.g., Fig. 2).
3. Relative comprehensive results, considering various datasets, baselines and metrics.

## Weakness
1. Only relatively small tabular datasets are used, which is not the most widely used case for DNNs.
2. Some details may need to be further clarified.

---

> ### Author Response · Authors · 2021-11-12
> **Thank you for your review**
>
> Thank you for your thoughtful review. We answer your questions in the order presented, and sometimes ask for further clarification.
>
> > How to compute the integral of SNS, what would be the complexity?
>
> We do not compute the integral in SNS exactly, but rather the integral is approximated by a summation over a set of points of a specified resolution--- in this paper, we use an interpolation of 10 points (see page 7). Approximating this path interval via a mesh grid is a common method in many explanation techniques, most notably Integrated Gradients  [Sundararajan et al., 2017]. The complexity of this calculation is linear in the number of points in the interpolation.
>
> > There is no causal model built and therefore, why the CFE in this paper is a counterfactual. If it is, which counterfactual distribution is it sampled from?
>
> In this paper, we use the term “counterfactual” to refer to a point $x’$ that causes a model to predict differently than a given reference point $x$. For example, a counterfactual example for some rejected applicant under a loan application model would be a similar application from another individual who the model decided to accept. This is distinct from how the term is used widely in the causality literature, where a counterfactual is given by an intervention on a causal model that is assumed to generate data observations. Our use of the term is consistent with usage in the explainability literature [Wachter et al., 2018; Ustun et al., 2019; Sharma et al., 2019; Poyiadzi et al., 2020; Pawelczyk et al., 2020a; Van Looveren & Klaise, 2019; Mahajan et al., 2019; Verma et al., 2020; Laugel et al., 2018; Keane & Smyth, 2020.].
>
> Counterfactuals are particularly popular explanations method in business [McGrath 2018] and legal [Wachter et al., 2018] applications, due to a number of desirable properties, including compliance with EU and US laws requiring explanations in sensitive contexts (e.g. ECOA, personal finance); their ability to generate explanations for a model of arbitrary complexity; the fact that counterfactual explanations release very little information about the underlying model, and that their creation can be fully automated [Barocas et al 2019].
>
> Thus, we make no claims that the counterfactual examples in this paper reflect interventions on a causal model for the data. We realize the overlapping terminology may be confusing for those more familiar with research surrounding causal models, and are happy to make the distinction between our focus and causal model based approaches in our introduction and related work.
>
>
> > What is a the x-axis and y-axis of Fig 2? I guess they are two input features but it is better to show them in the figure.
>
> Thank you for pointing this out. The x axis is the interpolation parameter $t$ for the computation of the integral in Definition 5: that is, $tx$ for $0 \leq t \leq 1$ creates a line from the baseline point (all zero input) to the point x for which a counterfactual is being generated. The y axis is the sigmoid output of the neural network over $tx$, which is what we are trying to maximize. This graph is meant to convey how the integral of the sigmoid output will be maximized via the SNS approach. We will update the draft with a more detailed explanation of the figure.
>
> >It is not very clear to me which terms are related to confidence in the equation of Theorem 2.
>
> The confidence of the model is its sigmoid output, i.e. $\sigma_w$.

---

> > ### Author Response · Authors · 2021-11-12
> > **Response part 2**
> >
> > >I did not fully get why small change in DI always means low invalidation. I guess this has to be shown formally.
> >
> > Section 4.2 explains this in more detail. We use distributional influence (DI) to capture relevant information about a model’s decision boundary, and analyze the differences between models resulting from variations in training (e.g., different initial randomizations) by studying the differences that arise in their decision boundaries. Distributional influence captures the relevant information about a model’s decision boundary, so if there is only a small change in distributional influence between models, then their decision boundaries will be similar in that region---meaning invalidation is less likely to occur.
> >
> >
> > >We know DNNs tend to be over-confident for unseen x. Would this result in any issue if we aim to find CFE with high confidence?
> >
> > Firstly, we note that, to generate a stable counterfactual with SNS,  we do not simply require that the counterfactual itself is confident, but all the points around it are confident as well---i.e. the entire path from the zero vector to the counterfactual point must be as confident as possible. If we were to search for a counterfactual that was only confident at the point itself,  it will not necessarily be stable to changes, which is often the case for unbounded counterfactual search and it is what the Fig 2 tries to motivate.
> >
> > Secondly, by “DNNs tend to be over-confident for unseen x” we assume the reviewer is talking about the case that DNNs can output pretty high confidences for off-manifold points and these counterfactuals might not be valid in practice. We notice that this can be the issue for all baseline approaches in the paper because the exact and closed-form description for the data manifold is often difficult to derive. SNS should be considered as a post-processing technique for the base counterfactual search algorithm, i.e. PGD, because SNS takes a counterfactual to begin with. That is, if the base counterfactual search algorithm comes with constraints that well describe the data manifold, it is natural to plug those constraints into SNS as well. Therefore, the underlying issue about off-manifold counterfactuals might be easier to solve if we improve the base search algorithm and pass that to SNS.
> >
> > Please let us know if this does not sufficiently answer your question.
> >
> > >What is the effect of relaxing the Lipschitz constant K from the penultimate output to the entire network? Would it increase or decrease, why?
> >
> > By “the Lipschitz constant of the entire network” we assume the reviewer is talking about the Lipschitz constant for each logit output. Consider the Lipschitz constant for the penultimate output vector as $K$ and an upper-bound of the Lipschitz constant for the logit output of class $j$ as $L_j$. A better way to describe the relation between $K$ and $L_j$ might be that $K * ||w_j||_q \leq L_j $ where $||w_j||$ is the norm of the $j$-th column of the last layer’s weight and $q=2$ in this paper. The reason the inequality holds is because the upper-bound of the Lipschitz Constant for the network’s output is given by the layer-wise product of Lipschitz constant for each layer (Gouk et al. 2021).
> >
> > To further help us address this question, could you please clarify what “it” refers to in the last sentence?
> >
> > >Only relatively small tabular datasets are used, which is not the most widely used case for DNNs.
> >
> > The datasets used in this paper are the ones most commonly used as benchmarks in counterfactual example evaluation [see e.g.  Ustun et al., 2019; Pawelczyk et al., 2020a, Rawal et al. 2021,, Verma et al., 2020]. Tabular data is the main target in this domain because it aligns with many of the use cases for counterfactual examples, such as credit scoring. We note that Taiwanese credit has ~30,000 rows; and HELOC has ~10,000, which we argue are sizable enough to create a reliable model on the (tabular, lower dimensional) data we target. The use case of our paper is for finding counterfactual examples in tabular environments where the individual input features have semantic meaning. We are happy to clarify this further in the paper.
> >
> > References (beyond those already in paper)
> >
> > McGrath, Rory, et al. "Interpretable credit application predictions with counterfactual explanations." arXiv preprint arXiv:1811.05245 (2018).
> > Equal Credit Opportunity Act. Bureau of Consumer Financial Protection. Nov 11 2021. URL: https://www.ftc.gov/enforcement/statutes/equal-credit-opportunity-act
> >
> > Gouk, H., Frank, E., Pfahringer, B., & Cree, M.J. (2021). Regularisation of neural networks by enforcing Lipschitz continuity. Machine Learning, 110, 393-416.

---

> > > ### Comment · Reviewer_ZP6K · 2021-11-26
> > > **Thanks for the reply!**
> > >
> > > Thanks for the authors' reply. Now I learned in counterfactual explanation the counterfactual is something completely different from what I was assuming. This is actually quite different from what I read in [1,2], but I can accept this difference.
> > >
> > > It is also great to learn from your explanation on the relationship between $K$ and $L_j$.
> > >
> > > [1] Yang, Fan, Sahan Suresh Alva, Jiahao Chen, and Xia Hu. "Model-Based Counterfactual Synthesizer for Interpretation." arXiv preprint arXiv:2106.08971 (2021).
> > > [2] Mahajan, Divyat, Chenhao Tan, and Amit Sharma. "Preserving causal constraints in counterfactual explanations for machine learning classifiers." arXiv preprint arXiv:1912.03277 (2019).

---

### Decision · Program_Chairs · 2022-01-20

**Decision:**

Accept (Poster)

**Comment:**

Most of the discussion centered around whether the underlying question in the literature is setup correctly in terms of its relationship to causality as the question being asked is one of an intervention. The underlying literature makes an attempt at not including things that can't be intervened on like age, but the setup of a "counterfactual" could benefit from a causal take.

Holding that aside, the paper makes progress on an established question though analysis that reveals that the Lipschitz continuity and confidence are important for causality and Stable Neighbor Search for generating counterfactuals.

The most negative reviewer in discussion writes that they're okay with the paper being accepted if the rest of the reviewers are positive. The rest of the reviewers are positive with one mentioning that the paper is well written and interesting in the discussion and that the author replies cleared up the issues about counterfactuals.